# PSEUDO PHYSICS-INFORMED NEURAL OPERATORS

## ABSTRACT

Recent advancements in operator learning are transforming the landscape of computational physics and engineering, especially alongside the rapidly evolving field of physics-informed machine learning. The convergence of these areas offers exciting opportunities for innovative research and applications. However, merging these two realms often demands deep expertise and explicit knowledge of physical systems, which may be challenging or even impractical in relatively complex applications. To address this limitation, we propose a novel framework: Pseudo Physics-Informed Neural Operator (PPI-NO). In this framework, we construct a surrogate physics system for the target system using partial differential equations (PDEs) derived from simple, rudimentary physics knowledge, such as basic differential operators. We then couple the surrogate system with the neural operator model, utilizing an alternating update and learning process to iteratively enhance the model's predictive power. While the physics derived via PPI-NO may not mirror the ground-truth underlying physical laws — hence the term "pseudo physics" — this approach significantly enhances the accuracy of current operator learning models, particularly in data scarce scenarios. Through extensive evaluations across five benchmark operator learning tasks and an application in fatigue modeling, PPI-NO consistently outperforms competing methods by a significant margin. The success of PPI-NO may introduce a new paradigm in physics-informed machine learning, one that requires minimal physics knowledge and opens the door to broader applications in data-driven physics learning and simulations.

## 1 Introduction

Operator learning, a dynamic and rapidly evolving domain, has seen remarkable advancements with the advent of neural operators. Rooted in the express power of neural networks, neural operators have transformed computational problem-solving methods. Prominent examples include Fourier Neural Operators (FNO) (Li et al., 2020c), Deep Operator Net (DONet) (Lu et al., 2021) and other frameworks such as (Cao, 2021; Hao et al., 2023). FNO employs Fourier transform for global convolution and function transformation, while DONet introduces two sub-networks — the branch net and trunk net — to extract representations from the functional space and query locations, respectively, enabling predictions akin to attention mechanisms (Vaswani et al., 2017).

For trading for model capacity and performance, neural operators often require a substantial amount of training data to perform optimally. This demand poses significant challenges, particularly in complex problems, where training data can be scarce and costly to acquire. In response, the field of physics-informed machine learning, including physics-informed neural networks (PINN) (Raissi et al., 2019), has shown promise by incorporating physical laws as soft constraints during training. This approach serves as a regularization technique, effectively embedding a fundamental understanding of physics into the model to lessen its reliance on extensive training data. Building on this idea, the concept of physics-informed neural operators (PINO) has emerged, with PINO integrating physical laws as soft constraints to enhance model fidelity while reducing data quantity. This approach has been used in (Wang et al., 2021; Li et al., 2021) for DONet and FNO training.

Despite the success of PINO, the necessity for a thorough understanding of the underlying physics can pose a significant hurdle, especially in complex applications such as in fracture mechanics and climate modeling. In those scenarios, the detailed physical knowledge is often unavailable or difficult to identify, and it is often prohibitively expensive to collect extensive data. These challenges can render the current methods unavailable or impractical.

To navigate these challenges while retaining the benefits of physics-informed learning, our work introduces the Pseudo Physics-Informed Neural Operator (PPI-NO). This framework bypasses the need for exhaustive physical comprehension by constructing a neural-network-based partial differential equation (PDE) that characterizes the target system directly from data. The neural PDE is then coupled with the neural operator for alternating updates and training, enabling iterative extraction, refinement and integration of physics knowledge to enhance operator learning. The contribution of this work lies in the following three aspects:

1. To our knowledge, PPI-NO is the first work to enhance standard operator learning pipeline using physics directly learned from *sparse data*, delivering superior accuracy without the need for in-depth physical understanding or extensive data collection.

2. The success of PPI-NO also opens up a new paradigm of physics-informed machine learning where only rudimentary physics assumptions (in this case, the basic differential operations) are required rather than in-depth or rigorous expert knowledge, extending the spectrum of the physics-informed learning for experts of different levels.

3. The effectiveness of PPI-NO is validated through extensive evaluations on five commonly used benchmark operator learning tasks in literature (Li et al., 2020c; Lu et al., 2022), including Darcy flow, nonlinear diffusion, Eikonal, Poisson and advection equations, as well as one application in fatigue modeling in fracture mechanics, where the ground-truth holistic PDE system is unknown.

## 2 Background

**Problem Formulation.** Operator learning seeks to approximate an operator that maps input parameters and/or functions to corresponding output functions. In most practical cases, operator learning rises in the context of solving partial differential equations (PDEs), where the operator corresponds to the solution operator of the PDE. Assume a PDE system:

$$\mathcal{N}[u](\mathbf{x}) = f(\mathbf{x}), \quad \mathbf{x} \in \Omega \times [0, \infty), \tag{1}$$

where $\mathbf{x}$ is a compact notation for the spatial and temporal coordinates, $\Omega$ is the spatial domain, $[0, \infty)$ is the temporal domain, $\mathcal{N}$ is a nonlinear differential operator, $u(\mathbf{x})$ is the solution function, and $f(\mathbf{x})$ is the source term. Solving the PDE system is to find the solution function $u(\mathbf{x})$ that satisfies the PDE system equation (1) as well as the initial and boundary conditions. This task often necessitates the use of computationally expensive numerical solvers such as finite element method (FEM) or finite difference method (FDM). To alleviate the computational challenge, we aim to learn the solution operator of the PDE system, $\psi : \mathbb{F} \to \mathbb{U}$ using a training dataset $\mathcal{D} = \{(\mathbf{f}_n, \mathbf{u}_n)\}_{n=1}^{N}$, which consists of discretized functions $u(\cdot)$ and $f(\cdot)$ at a set of collocations points. Once the operator model is trained, it can be used to directly predict the solution function $u$ for new instances of the input $f$, offering a much more efficient alternative to running numerical solvers from scratch. However, the training dataset still needs to be generated offline using numerical solvers.

**Fourier Neural Operator (FNO)** (Li et al., 2020c) represents a significant leap in neural network architecture for operator learning, especially in solving PDEs. For a given discretized input function $\mathbf{f}$, FNO first employs a feed-forward network (FFN) on each component of $\mathbf{f}$ at its respective sampling location, thereby lifting the input into a higher-dimensional channel space. The core of FNO is the Fourier layer, which performs a linear transformation followed by a nonlinear activation within the functional space, $h(\mathbf{x}) \leftarrow \sigma \left( \mathcal{W}h(\mathbf{x}) + \int \kappa(\mathbf{x} - \mathbf{x}')h(\mathbf{x}')d\mathbf{x}' \right)$, where $h(\mathbf{x})$ is the input to the Fourier layer, $\kappa(\cdot)$ the integration kernel, and $\sigma(\cdot)$ the activation function. The convolution operation in this context is efficiently computed using the convolution theorem: $\int \kappa(\mathbf{x} - \mathbf{x}')h(\mathbf{x}')d\mathbf{x}' = \mathcal{F}^{-1}[\mathcal{F}[\kappa] \cdot \mathcal{F}[h]](\mathbf{x})$, where $\mathcal{F}$ and $\mathcal{F}^{-1}$ denote the Fourier and inverse Fourier transforms, respectively. The Fourier layer's efficiency stems from performing the Fast Fourier Transform (FFT) on $h$, multiplying it with the discretized kernel in the frequency domain, and then applying the inverse FFT. The local linear transformation, $\mathcal{W}h(\mathbf{x})$, is executed through conventional convolution operations. After multiple Fourier layers, the final output is obtained by the channel-wise application of another FFN, projecting the representation back to the original space.

**Deep Operator Network (DONet)** (Lu et al., 2021) is another prominent work in operator learning. The architecture of a DONet is structured into two primary components: the branch network and the trunk network, learning representations for the input functions and querying locations, respectively.

Consider an input function $f(\mathbf{x}) \in \mathbb{F}$ evaluated at $m$ sensor locations $\{\mathbf{x}_1, \mathbf{x}_2, \cdots, \mathbf{x}_m\}$ and an output function $u \in \mathbb{U}$. The branch network receives the values $[f(\mathbf{x}_1), f(\mathbf{x}_2), \cdots, f(\mathbf{x}_m)]$ and outputs a feature representation $[b_1, b_2, \cdots, b_p]^\top \in \mathbb{R}^p$. Concurrently, the trunk network processes a querying location $\mathbf{x}$ and outputs another feature vector $[t_1, t_2, \cdots, t_p]^\top \in \mathbb{R}^p$. The approximation of the output function $u(\mathbf{x})$ is computed as a sum of products of the corresponding elements from the branch and trunk networks, $\mathcal{G}[f](\mathbf{x}) \approx \sum_{k=1}^p b_k t_k$, where $\mathcal{G}$ is the learned operator mapping input function $f$ to the corresponding output function $u$.

**Physics-Informed Neural Operator (PINO)** (Wang et al., 2021; Li et al., 2021) has recently emerged as a promising approach to address the data scarcity issue in operator learning. PINO embeds physical laws — typically governing equations — into the learning process. The incorporation of physical principles not only enhances the model's adherence to ground-truth phenomena but also reduces its dependency on extensive training data. Mathematically, the integration of physics into the learning process can be viewed as an additional regularization term in the loss function. Let $\mathcal{L}_{\text{data}}$ represent the standard data-fitting loss term (*e.g.,* the mean squared error between the predicted and actual outputs), the physics-informed term $\mathcal{L}_{\text{physics}}$ can be the residual of the governing PDEs evaluated at the neural network's outputs. The total loss function $\mathcal{L}$ for a PINO model is then expressed as

$$\mathcal{L} = \mathcal{L}_{\text{data}} + \lambda \mathcal{L}_{\text{physics}},$$

where $\lambda$ is a weighting factor that balances the importance of data-fitting versus physics compliance. This approach encourages the model to learn solutions that are not only consistent with the provided data but also physically plausible.

## 3 Methodology

In the absence of the underlying physics knowledge (*i.e.,* the PDE system (1) is not available), it is impossible to construct the physics loss term in the PINO framework. To address this challenge, we propose a "pseudo" physics-informed operator learning framework motivated by the need to uncover the underlying physical laws using available data. This approach is particularly useful in relatively more complex applications, where data is often costly or sparse while the underlying physics is hard to fully understand. Our model architecture is depicted in Figure 1.

### 3.1 Pseudo Physics System Learning

As the first step, we propose a novel approach to learn the physics system using scarce training data. Our key observation is that, *although the mapping from $f$ to $u$ can be intricate and may necessitate information across the entire domain (in theory, $u$ is an integration of the Green's function multiplied with $f$ over the domain), the underlying PDE system* (1) *simplifies to a local combination of $u$ and its derivatives.* We therefore use a neural network $\phi$ to approximate the general form of $\mathcal{N}$,

$$\mathcal{N}[u](\mathbf{x}) \approx \phi\left(\mathbf{x}, u(\mathbf{x}), S_1(u)(\mathbf{x}), \ldots, S_Q(u)(\mathbf{x})\right), \qquad (2)$$

where $\{S_j\}_{j=1}^Q$ are $Q$ derivative operators that we believe should be present in the system, such as $\partial_t u, \partial_{tt} u, \partial_{x_1} u, \partial_{x_2} u, \partial_{x_1 x_1} u, \partial_{x_1 x_2} u, \partial_{x_2 x_2} u$, and more.

The inherent local combination nature of the PDE representation decouples the values of $u$ and its derivatives across various sampling locations, thereby significantly increasing the number of available training data points. For instance, consider sampling the input function $f$ and output function $u$ on a $128 \times 128$ grid. A single pair of discretized input and output functions, denoted as $(\mathbf{f}, \mathbf{u})$, is typically insufficient for a neural operator to effectively learn the mapping $f \rightarrow u$. However, this sample can be decomposed into $128 \times 128 = 16,384$ training data points across various (spatial and temporal) locations to train $\phi$ as outlined in (2). Hence, even with a small number of $(\mathbf{f}, \mathbf{u})$ pairs for operator learning, the learning of the PDE system $\mathcal{N}$ via our formulation in (2) can still achieve accuracy, thanks to the much greater number of training data points that can be derived from these pairs.

We use an $L_2$ loss to estimate the parameters of $\phi$, which is defined as

$$\mathcal{L}_\phi = \sum_{n=1}^N \sum_{j=1}^M \left[\phi(\mathbf{x}_j, u_n(\mathbf{x}_j), S_1(u_n)(\mathbf{x}_j), \ldots, S_Q(u_n)(\mathbf{x}_j)) - f_n(\mathbf{x}_j)\right]^2, \qquad (3)$$

where $f_n(\cdot)$ and $u_n(\cdot)$ are the input and output functions in $n$-th training example, and $\{\mathbf{x}_1, \ldots \mathbf{x}_M\}$ are the locations at which we discretize $f_n$ and $u_n$.

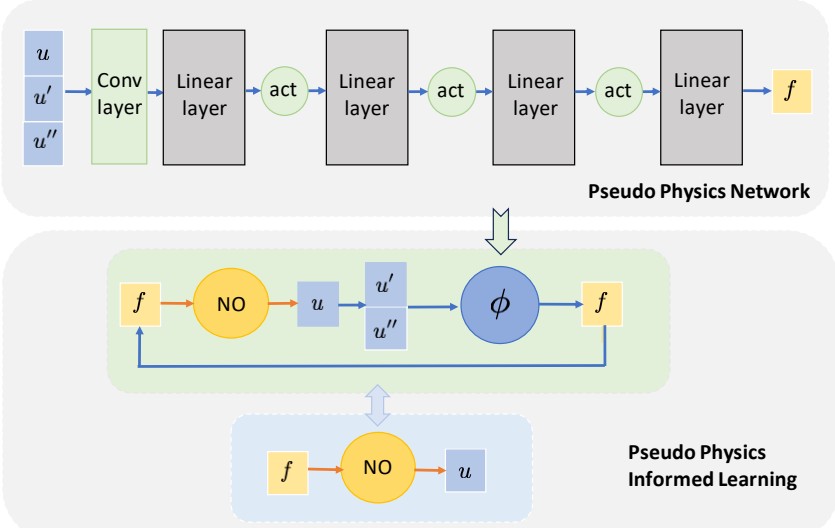

Figure 1: The illustration of the Pseudo Physics-Informed Neural Operator (PPI-NO). At the top, a black-box PDE representation is learned through the neural network $\phi$. At the bottom, the acquired "pseudo" physics laws are utilized to form a reconstruction loss, thereby regulating the NO training.

We use numerical difference to obtain the derivatives of each $u_n$, namely, $S_k(u_n)$ ($1 \leq k \leq Q$), and then feed these inputs to the neural network $\phi$ to compute the prediction. As the numerical difference method may introduce errors when calculating derivatives, we incorporate a convolution layer in $\phi$ to collect and integrate neighborhood information about $u$ and its numerical derivatives, aiming to compensate for these errors. After that, we use feed-forward layers to sequentially perform linear transform and nonlinear activation to obtain the prediction at each sampling location; see Fig. 1 top. The learned neural network mapping $\phi : u \rightarrow f$, although black-box in nature, can encapsulate valuable physics knowledge inherent in the data employed for operator learning.

Our method can be easily adapted to scenarios where the input and output functions are irregularly sampled, and numerical differentiation is no longer applicable. In such cases, we can employ smooth function estimators, such as kernel interpolation (Long et al., 2024) or Bayesian B-splines (Sun et al., 2022), to estimate the gradient information from data. These gradient estimations are then fed into our PDE neural network $\phi$ for further learning.

### 3.2 Coupling Neural Operator with Pseudo Physics

Next, we leverage the pseudo physics laws embedded in the learned mapping $\phi : u \rightarrow f$ to enhance the neural operator learning process. Specifically, we use $\phi$ to reconstruct $f$ from the $u$ predicted by the neural operator. In this way, our approach goes beyond relying solely on the training data; it uses the physics learned in the previous step to incorporate a reconstruction error into optimizing the neural operator parameters.

Initially, we train the neural operator $\psi : f \rightarrow u$ using the available training data, creating a preliminary model. This model is developed using FNO or DONet or other neural operators. The focus is to first establish a basic understanding of the relationship between $f$ and $u$ from the limited data. Next, the loss function for $\psi$ is augmented using the physics laws learned in the first step,

$$\mathcal{L} = \sum\nolimits_{n=1}^{N} \mathcal{L}_2(\psi(f_n), u_n) + \lambda \cdot \mathbb{E}_{p(f')} \left[ \mathcal{L}_2(f', \phi(\psi(f'))) \right], \tag{4}$$

where the first term is the $\mathcal{L}_2$ loss for data fitting (as in the standard neural operator training), and the second term is the expected reconstruction error for the input function. The second term incorporates the physics laws embedded in $\phi(\cdot)$, and $\lambda$ is a weight factor that balances the training data loss against the reconstruction error.

In practice, the expected reconstruction error does not have a closed form. One can sample a collection of $f'$ from the underlying distribution of the input function $p(\cdot)$, *e.g.,* a Gauss random field

or Gaussian process, and then employ a Monte-Carlo approximation,

$$\mathcal{L} = \sum_{n=1}^{N} \mathcal{L}_2(\psi(f_n), u_n) + \lambda \frac{1}{N'} \sum_{n=1}^{N'} \mathcal{L}_2(f'_n, \phi(\psi(f'_n))), \tag{5}$$

where $N'$ is the number of input function samples.

To enhance the operator learning process, the model is iteratively refined. In each iteration, we first fine-tune the neural operator $\psi$ with the pseudo physics $\phi$ fixed, and then fix $\psi$, fine-tune $\phi$ to refine the physics representation. This fine-tuning loop is carried out for multiple iterations, allowing for continuous improvement of the neural operator based on the refined physics.

This methodology mirrors human experts' approach to physics system modeling, where sparse data is used to learn the physics laws inspired by simple differential operation (up to the 2nd order to imitate human experts), and then these laws are utilized to generalize the system for data generation. The reconstruction loss term augments the operator learning with additional information, leading to potential improvement upon only training with sparse data.

## 4 Related Work

Neural operator learning is expanding rapidly. In addition to FNO (Li et al., 2022) and DONet (Lu et al., 2021), notable works include the Low-rank Neural Operator (LNO) introduced by Li et al. (2020c), employing low-rank structures to approximate the integration. The Graph Neural Operator (GNO) (Li et al., 2020a) integrates Nystrom approximation with graph neural networks, while the Multipole Graph Neural Operator (MGNO) by the same authors (Li et al., 2020b) leverages multiscale kernel decomposition. Gupta et al. (2021) contributed with multiwavelet transformations for the operator's kernel. Lu et al. (2022) proposed POD-DONet to enhance the stability of DONet by replacing the trunk net with POD bases constructed from data. Another DONet variant by Seidman et al. (2022) used an FFN to combine the outputs of the branch net and trunk net for prediction. A line of efforts attempted to build neural operators via transformer architectures, such as (Cao, 2021; Hao et al., 2023) Recently, Kovachki et al. (2023) provided a comprehensive review of neural operators. There are also recent advances in kernel operator learning strategies made by Long et al. (2022) and Batlle et al. (2023).

Physics-Informed Neural Networks (PINNs) (Raissi et al., 2019) mark a significant advancement in scientific machine learning. PINNs integrate physical laws directly into the learning process, making them effective for solving differential equations and understanding complex physical systems. This methodology is particularly beneficial in scenarios where data is sparse or expensive to obtain. Pioneering the concept of PINO, Li et al. (2021) introduced a dual-resolution approach that combines low-resolution empirical data with high-resolution PDE constraints. This method achieves precise emulation of solution operators across various PDE classes. In parallel, physics-informed DONet by Wang et al. (2021) incorporate regularization strategies enforcing physical law adherence into the training of DONets. Zanardi et al. (2023) presented an approach using PINO for simulations in non-equilibrium reacting flows. Lee et al. (2023) proposed opPINN, a framework combining physics-informed neural networks with operator learning for solving the Fokker-Planck-Landau (FPL) equation. Rosofsky et al. (2023) provided a review of applications of physics-informed neural operators. However, existing methods demand one should know the physics laws beforehand, which might not be feasible in many practical applications or complex systems. Our method offers a simple and effective framework, enabling the extraction of implicit physics laws directly from data, even when the data is sparse. Empirically, these pseudo physics laws have proven to be highly beneficial in enhancing the performance of operator learning, as demonstrated in Section 5.

Our work is also related to the cycle consistence framework (Zhu et al., 2017) for image-to-image translation. A critical difference is that cycle-consistence performs *unpaired* image-to-image translation, while our method aims for accurate paired translation (mapping). In cycle-consistence, the translation is viewed successfully as long as the translated images follow some target distribution. Hence, cycle-consistence has a much more relaxed objective. Another key difference is that our method aims to improve the learning of a function-to-function mapping with very limited data— that is why we first learn a "pseudo physics" representation. The cycle-consistence relies on adversarial training which typically requires a large amount of data to obtain successful learning outcomes.

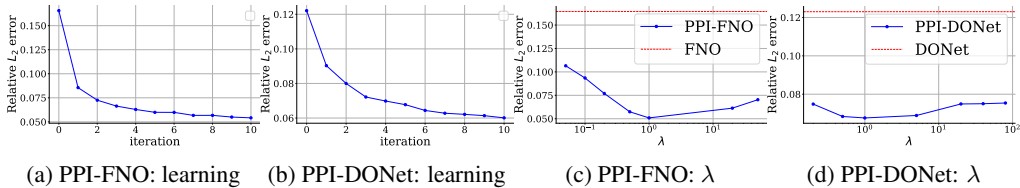

| (a) PPI-FNO: learning | (b) PPI-DONet: learning | (c) PPI-FNO: $\lambda$ | (d) PPI-DONet: $\lambda$ |

Figure 2: Learning curve of PPI-FNO on Darcy Flow (a), and of PPI-DONet on nonlinear diffusion (b). In (c) and (d) we show how the weight $\lambda$ of "pseudo physics" affects the operator learning performance. The horizontal line in (c) and (d) are the relative $L_2$ errors of standard FNO and DONet.

## 5 Experiments

**Dataset.** We tested on five commonly used benchmark operator learning problems in literature (Li et al., 2020c; Lu et al., 2022), including *Darcy Flow*, *Nonlinear Diffusion*, *Eikonal*, *Poisson* and *Advection*. In addition, we examined our method in an application in fatigue modeling. The task is to predict the stress intensity factor (SIF) for semi-elliptical surface cracks on plates, given three geometric parameters that characterize the cracks (Merrell et al., 2024); see Appendix Fig. 4. The SIF plays a critical role in modeling crack growth by quantifying the stress state near the tip of a crack, and hence SIF computation and analysis is extremely important in fatigue modeling and fracture mechanics (Anderson and Anderson, 2005). The SIF computation is expensive, because it typically needs to run finite element method (FEM) or extended FEM with very fine meshes (Kuna, 2013). Due to the complex sequence of computational steps involved in SIF calculation, there is no holistic PDE that directly models the relationship between the geometric features and the SIF function. Instead, SIF computation typically relies on numerical methods and the extraction of local stress fields near the crack tip. The details about all the dataset are given in Section A of the Appendix.

**Method and Settings.** We evaluated our method based on two popular NO models, FNO and DONet. For learning the pseudo physics laws via the neural network $\phi$ — see (2) — we tuned the kernel size from $\{(3, 3), (5, 5), (7, 7), (9, 9)\}$. The stride was set to 1 and padding was set to "same" to ensure the output shape does not change. In the subsequent FFN, we chose the number of layers from $\{3, 4, 5, 6\}$, and the layer width from $\{16, 32, 64\}$. We used GeLU activation. For the cases of Darcy Flow, Eikonal and Poisson, we used the following derivatives $\{\partial_{x_1} u, \partial_{x_2} u, \partial_{x_1 x_1} u, \partial_{x_2 x_2} u, \partial_{x_1 x_2} u\}$, and for the other cases, we used $\{\partial_x u, \partial_{xx} u, \partial_t u, \partial_{tt} u, \partial_{xt} u\}$. Since SIF is a 1d function (the input is the angle), we used the derivatives $\{\partial_x u, \partial_{xx} u\}$. For FNO, we set the number of modes to 12 and channels to 32 (in the lifted space). We varied the number of Fourier layers from $\{2, 3, 4\}$. For DONet, in all the cases except Darcy Flow, the trunk net and branch net were constructed as FFNs. We varied the number of layers from $\{2, 3, 4\}$ and the layer width was chosen from $\{30, 40, 50, 60\}$, with ReLU activation. For the case of Darcy flow, we found that DONet with only feed-forward layers exhibited inferior performance. To address this, we introduced convolution layers into the branch net. We selected the number of convolution layers from $\{3, 5, 7\}$, and employed batch normalization and leaky ReLU after each convolution layer. To incorporate the learned pseudo physics representation into the training of FNO or DONet, we randomly sampled 200 input functions to construct the second loss term in (5). We set the maximum number of iterations to 10 and selected the weight $\lambda$ from $[10^{-1}, 10^2]$. All the models were implemented by PyTorch (Paszke et al., 2019), and optimized with ADAM (Kingma and Ba, 2014). The learning rate was selected from $\{10^{-4}, 5 \times 10^{-4}, 10^{-3}\}$. The number of epochs for training or fine-tuning FNO, DONet and pseudo physics network $\phi$ was set to 500 to ensure convergence. For each operator learning benchmark, we simulated 100 examples for testing, and varied the number of training examples from $\{5, 10, 20, 30\}$, except for Advection, we ran with $\{20, 30, 50, 80\}$ training examples. For SIF prediction (which is much more challenging), we experimented with training size from $\{400, 500, 600\}$, and employed 100 test examples. We repeated the evaluation for five times, each time we randomly sampled a different training set. We ran experiments on workstations equipped with Nvidia Geforce RTX 4090 and Intel I9 CPU.

### 5.1 Results and Analysis

**Predictive performance.** We reported the average relative $L_2$ error and the standard deviation (before and after incorporating the pseudo physics laws) in Table 1 and Table 2. The model trained with

| Training size | 5 | 10 | 20 | 30 |
|---|---|---|---|---|
| FNO | $0.4915 \pm 0.0210$ | $0.3870 \pm 0.0118$ | $0.2783 \pm 0.0212$ | $0.1645 \pm 0.0071$ |
| PPI-FNO | $0.1716 \pm 0.0048$ | $0.0956 \pm 0.0084$ | $0.0680 \pm 0.0031$ | $0.0642 \pm 0.0010$ |
| Error Reduction | 65.08% | 75.29% | 75.56% | 60.97% |
| DONet | $0.8678 \pm 0.0089$ | $0.6854 \pm 0.0363$ | $0.5841 \pm 0.0279$ | $0.5672 \pm 0.0172$ |
| PPI-DONet | $0.5214 \pm 0.0543$ | $0.3408 \pm 0.0209$ | $0.2775 \pm 0.0224$ | $0.2611 \pm 0.0084$ |
| Error Reduction | 39.91% | 50.27% | 52.49% | 53.96% |

(a) *Darcy flow*

| Training size | 5 | 10 | 20 | 30 |
|---|---|---|---|---|
| FNO | $0.2004 \pm 0.0083$ | $0.1242 \pm 0.0046$ | $0.0876 \pm 0.0061$ | $0.0551 \pm 0.0021$ |
| PPI-FNO | $0.0105 \pm 0.0016$ | $0.0066 \pm 0.00023$ | $0.0049 \pm 0.00037$ | $0.0038 \pm 0.00039$ |
| Error Reduction | 94.76% | 94.68% | 94.40% | 93.10% |
| DONet | $0.3010 \pm 0.0119$ | $0.2505 \pm 0.0057$ | $0.1726 \pm 0.0076$ | $0.1430 \pm 0.0036$ |
| PPI-DONet | $0.1478 \pm 0.0126$ | $0.1161 \pm 0.0124$ | $0.1032 \pm 0.0059$ | $0.0842 \pm 0.0041$ |
| Error Reduction | 50.89% | 53.65% | 40.20 % | 41.11% |

(b) *Nonlinear diffusion*

| Training size | 5 | 10 | 20 | 30 |
|---|---|---|---|---|
| FNO | $0.2102 \pm 0.0133$ | $0.1562 \pm 0.0098$ | $0.0981 \pm 0.0022$ | $0.0843 \pm 0.0020$ |
| PPI-FNO | $0.0678 \pm 0.0026$ | $0.0582 \pm 0.0043$ | $0.0493 \pm 0.0023$ | $0.0459 \pm 0.0010$ |
| Error Reduction | 67.74% | 62.74% | 49.74% | 45.55% |
| DONet | $0.3374 \pm 0.0944$ | $0.1759 \pm 0.0065$ | $0.1191 \pm 0.0047$ | $0.1096 \pm 0.0037$ |
| PPI-DONet | $0.1302 \pm 0.0127$ | $0.0907 \pm 0.0093$ | $0.0714 \pm 0.0011$ | $0.0700 \pm 0.0007$ |
| Error Reduction | 61.41% | 48.43% | 40.05% | 36.13% |

(c) *Eikonal*

| Training size | 5 | 10 | 20 | 30 |
|---|---|---|---|---|
| FNO | $0.2340 \pm 0.0083$ | $0.1390 \pm 0.0007$ | $0.0895 \pm 0.0008$ | $0.0698 \pm 0.0014$ |
| PPI-FNO | $0.1437 \pm 0.0062$ | $0.0771 \pm 0.0018$ | $0.0544 \pm 0.0009$ | $0.0458 \pm 0.0003$ |
| Error Reduction | 38.59% | 44.53% | 39.22% | 34.38% |
| DONet | $0.6142 \pm 0.0046$ | $0.5839 \pm 0.0090$ | $0.5320 \pm 0.0028$ | $0.5195 \pm 0.0040$ |
| PPI-DONet | $0.5275 \pm 0.0037$ | $0.5001 \pm 0.0042$ | $0.4450 \pm 0.0010$ | $0.4258 \pm 0.0040$ |
| Error Reduction | 14.12% | 14.35% | 16.35% | 18.04% |

(d) *Poisson*

| Training size | 20 | 30 | 50 | 80 |
|---|---|---|---|---|
| FNO | $0.4872 \pm 0.0097$ | $0.4035 \pm 0.0086$ | $0.3019 \pm 0.0085$ | $0.2482 \pm 0.0059$ |
| PPI-FNO | $0.3693 \pm 0.0099$ | $0.3224 \pm 0.0123$ | $0.2236 \pm 0.0075$ | $0.1698 \pm 0.0075$ |
| Error Reduction | 24.20% | 20.10% | 25.94% | 31.59% |
| DONet | $0.5795 \pm 0.0045$ | $0.4810 \pm 0.0092$ | $0.3882 \pm 0.0086$ | $0.3164 \pm 0.0072$ |
| PPI-DONet | $0.3630 \pm 0.0112$ | $0.2897 \pm 0.0097$ | $0.2629 \pm 0.0053$ | $0.2120 \pm 0.0065$ |
| Error Reduction | 37.36% | 39.77% | 32.28% | 33.00% |

(e) *Advection*

Table 1: Relative $L_2$ error in five operator learning benchmarks, where "PPI" is short for Pseudo-Physics Informed". The results were averaged from five runs.

| Training size | 400 | 500 | 600 |
|---|---|---|---|
| FNO | $0.1776 \pm 0.0150$ | $0.1695 \pm 0.0090$ | $0.1122 \pm 0.0094$ |
| PPI-FNO | $0.1166 \pm 0.0064$ | $0.1151 \pm 0.0093$ | $0.0850 \pm 0.0060$ |
| Error Reduction | 34.35% | 32.09% | 24.24% |
| DONet | $0.5318 \pm 0.0095$ | $0.5155 \pm 0.0200$ | $0.4037 \pm 0.0331$ |
| PPI-DONet | $0.3490 \pm 0.0034$ | $0.3468 \pm 0.0074$ | $0.3299 \pm 0.0066$ |
| Error Reduction | 34.37% | 32.73% | 18.28% |

Table 2: SIF prediction error for plate surface cracks in fatigue modeling.

the learned physics laws (see (5)) is denoted as PPI-FNO or PPI-DONet, short for Pseudo Physics Informed FNO/DONet. In all the cases, with our pseudo physics informed approach , the prediction error of both FNO and DONet experiences a large reduction. For instance, across all training sizes in *Darcy Flow* and *nonlinear diffusion*, PPI-FNO reduces the relative $L_2$ error of the ordinary FNO by

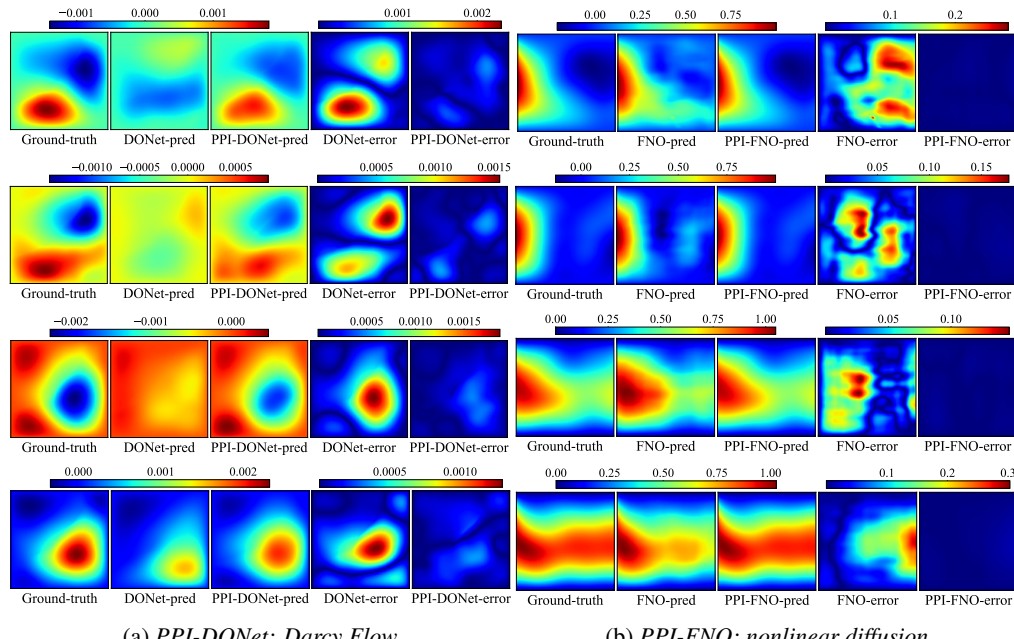

(a) *PPI-DONet: Darcy Flow*  (b) *PPI-FNO: nonlinear diffusion*

Figure 3: Examples of the prediction and point-wise error of PPI-DONet and PPI-FNO on *Darcy Flow* and *nonlinear diffusion*, respectively. From top to bottom, the models were trained with 5, 10, 20, 30 examples.

over 60% and 93%, respectively. In *Darcy Flow* with training sizes 10 to 30, PPI-DONet reduces the error of the ordinary DONet by over 50%. In SIF prediction, our method applied to both FNO and DONet reduced the error by over 30% with training size 400 and 500. Even the minimum reduction across all cases achieves 14.12% (PPI-DONet over DONet on *Poisson* with training size 5).

Together these results demonstrate the strong positive impact of the learned physics by our neural network model $\phi$ specified in Section 3.1. Although it remains opaque and non-interpretable, it encapsulates valuable knowledge that greatly enhances the performance of operator learning, in particular with limited data.

Next, we assessed the accuracy of the learned physics laws by examining the relative $L_2$ error in predicting the source functions $f$ from $\phi$ (see (2)). We tested on *Darcy Flow*, *nonlinear diffusion*, and *Eikonal*. We compared a baseline method that removes the convolution layer of $\phi$, leaving only the feed-forward layers. The average relative $L_2$ error and standard deviation are reported in Table 3.

It can be observed that in nearly every case, adding a convolution layer indeed significantly improves the accuracy of $\phi$. This improvement might be attributed to the convolution layer's ability to integrate neighboring information and compensate for the error introduced by the numerical difference in approximating the derivatives. We also experimented with multiple convolution layers, but the improvement was found to be marginal.

In addition, we also found the operator learning improvement

| Benchmark | FFN | Ours |
|---|---|---|
| Darcy Flow | 0.1819±0.0026 | **0.1392± 0.0080** |
| Nonlinear Diffusion | 0.0660±0.0069 | **0.0233±0.0005** |
| Eikonal | 0.0144±0.0009 | **0.0108 ± 0.0006** |

(a) Training size=10

| Benchmark | FFN | Ours |
|---|---|---|
| Darcy Flow | 0.1413±0.0013 | **0.0688± 0.0032** |
| Nonlinear Diffusion | 0.0463±0.0022 | **0.0163±0.0002** |
| Eikonal | 0.0070±0.00005 | **0.0052 ± 0.0002** |

(b) Training size=30

Table 3: Relative $L_2$ error of using the learned back-box PDE network (2) to predict the input function $f$.

is relatively *robust* to the accuracy of our physics representation $\psi$. For instance, on *Darcy Flow* with training size 5 and 10, the relative $L_2$ error of $\phi$ network is 0.2285 and 0.1392, which is significantly bigger than with training size 30 where the relative $L_2$ error is 0.0688. Yet the error reduction upon

FNO (see Table 1a) under all the three training sizes is above 60%. The error reduction upon DONet is 40% for training size 5 and over 50% for training size 10 and 30. The results imply that even roughly capturing the underlying physics (with $\phi$) can substantially boost the operator learning performance.

**Point-wise prediction and point-wise error.** For a detailed assessment, we conducted a fine-grained evaluation by visualizing the predictions and point-wise errors made by each method. In Fig. 3a and 3b, we showcased the predictions and point-wise errors using PPI-DONet for *Darcy Flow*, PPI-FNO for *nonlinear diffusion*, respectively. Additional examples of predictions and point-wise errors are provided in Fig. 5, 7a, 7b, 8a, and 8b in the Appendix.

It is evident that without the assistance of the pseudo physics laws learned by our method, the ordinary DONet and FNO frequently missed crucial local structures, sometimes even learning entirely incorrect structures. For example, In Fig. 3a the first row, DONet missed one mode, while in the second and third row of Fig. 3a, DONet failed to capture all the local modes. After incorporating the learned physics, DONet (now denoted as PPI-DONet; see the third column) successfully captures all the local modes, including their shapes and positions. Although not all the details are exactly recovered, the point-wise error is substantially reduced, particularly in those high error regions of the ordinary DONet; see the fourth column of Fig. 3a. In another instance, as shown in Fig. 3b, where the ordinary FNO (second column) captured the global shape of the solution, but the mis-specification of many local details led to large point-wise errors across many regions (fourth column). In contrast, PPI-FNO (third column) not only identified the structures within the solution but also successfully recovered the details. As a result, the point-wise error (fifth column) was close to zero everywhere. Additional instances can be found in Fig. 7a, the first three rows illustrate that ordinary FNO (trained with 5, 10, and 20 examples, respectively) estimates an entirely incorrect structure of the solution, indicating that the training data is insufficient for FNO to capture even the basic structure of the solution. In contrast, after fine-tuning with our learned physics laws from the same sparse data, PPI-FNO accurately figured out the solution structures and yielded a substantial reduction in point-wise error across nearly everywhere. The point-wise error became uniformly close to zero. With 30 examples, the ordinary FNO was then able to capture the global structure of the solution, but the details in the bottom left, bottom right, and top right corners were incorrectly predicted. In comparison, PPI-FNO further recovered these details accurately.

Collectively, these results demonstrate that the pseudo physics extracted by our method not only dramatically boosts the overall prediction accuracy but also better recovers the local structures and details of the solution.

**Learning Behavior.** We examined the learning behavior of our method, which conducts an iterative, alternatingly fine-tuning process. We employed one *Darcy Flow*, one *nonlinear diffusion* and one *Eikonal* dataset, each with 30 examples. We show the test relative $L_2$ error along with the iterations in Fig. 2a, 2b, and Appendix Fig. 6a and Fig. 6b. As we can see, the predictive performance of our algorithm kept improving and tended to converge at last, affirming the efficacy of learning process.

**Ablation study on the PDE network $\phi$.** To confirm the efficacy of our designed PDE network $\phi$ in facilitating operator learning, we considered alternative designs for $\phi$: (1) using standard FNO to predict $f$ directly from $u$; no derivative information is included in the input; (2) removing the convolution layer in our model, and just keeping the FNN layers; the input is the same as our model, *i.e.,* the derivative information is included in the input. With different designs of $\phi$, we evaluated the PPI learning performance on the Darcy Flow benchmark. The relative $L_2$ error in predicting $f$ via $\phi$ and predicting $u$ is reported in Table 4. Our design of $\phi$ consistently outperforms alternative architectures by a notable margin, showing the effectiveness of learning a (black-box) PDE representation and improving the operator learning.

**Ablation study on the choice of derivatives.** We further investigated the PPI learning performance with respect to the choice of derivatives used in the PDE network. Specifically, we tested PPI-FNO on the Darcy-flow benchmark and varied the order of derivatives up to 0, 1, 2, and 3. The performance is reported in Table 5. We can see that although the accuracy of $\phi$ with derivatives up to the third order is slightly better than with derivatives up to the second order, the best operator learning performance was still achieved using derivatives up to the second order (which was used in our evaluations). This might be because higher-order derivative information can cause overfitting in the PDE network $\phi$ to a certain degree. Such higher-order information may not be critical to the actual mechanism of the physical system and can therefore impede the improvement of operator learning performance.

| Training size | 5 | 10 | 20 | 30 |
|---|---|---|---|---|
| FNO | 0.7229±0.0318 | 0.5759± 0.0126 | 0.4257± 0.0106 | 0.3160± 0.0037 |
| MLP | 0.7169±0.0160 | 0.6598± 0.0056 | 0.6464± 0.0029 | 0.6277± 0.0032 |
| Ours | **0.2285 ± 0.0147** | **0.1392 ± 0.0080** | **0.0898 ± 0.0046** | **0.0688 ± 0.0032** |

(a) Predicting $f$ via $\phi$ with different architectures.

| Training size | 5 | 10 | 20 | 30 |
|---|---|---|---|---|
| PPI-FNO with FNO as $\phi$ | 0.5853±0.0153 | 0.3871± 0.0124 | 0.2613± 0.0190 | 0.1629± 0.0064 |
| PPI-FNO with MLP as $\phi$ | 0.7262±0.0920 | 0.5516± 0.0699 | 0.4568± 0.0857 | 0.3983± 0.1051 |
| Standard FNO | 0.4915 ± 0.0210 | 0.3870 ± 0.0118 | 0.2783 ± 0.0212 | 0.1645 ± 0.0071 |
| Ours | **0.1716 ± 0.0048** | **0.0956 ± 0.0084** | **0.0680 ± 0.0031** | **0.0642 ± 0.0010** |

(b) Predicting $u$.

Table 4: The relative $L_2$ error with using different architectures of $\phi$ in pseudo-physics-informed (PPI) learning on Darcy Flow benchmark.

| Training size | 5 | 10 | 20 | 30 |
|---|---|---|---|---|
| order 0 | 0.7126±0.0131 | 0.5733±0.0208 | 0.4812±0.0399 | 0.3445±0.0182 |
| order ≤ 1 | 0.2926±0.0118 | 0.2006±0.0047 | 0.1379±0.0051 | 0.1084±0.0053 |
| order ≤ 2 | 0.2285±0.0147 | 0.1392±0.0080 | 0.0898±0.0046 | 0.0688±0.0032 |
| order ≤ 3 | **0.2058±0.0192** | **0.1123±0.0039** | **0.0712±0.0021** | **0.0585±0.0030** |

(a) Predicting $f$ via $\phi$.

| Training size | 5 | 10 | 20 | 30 |
|---|---|---|---|---|
| order 0 | 0.6352±0.0673 | 0.4523±0.0621 | 0.3570±0.0658 | 0.2737±0.0643 |
| order ≤ 1 | 0.3386±0.0259 | 0.2161±0.0083 | 0.1645±0.0114 | 0.1197±0.0132 |
| order ≤ 2 | **0.1716±0.0048** | **0.0956±0.0084** | **0.0680±0.0031** | **0.0642±0.0010** |
| order ≤ 3 | 0.2959±0.0381 | 0.1719±0.0213 | 0.1193±0.0158 | 0.0828±0.0054 |

(b) Predicting $u$.

Table 5: The relative $L_2$ error of PPI learning by incorporating different orders of derivatives. During the comparison with other operator learning methods, we used derivative orders up to 2 to run our method.

**Ablation study on the weight $\lambda$.** We examined the effect of the weight $\lambda$ of our "pseudo physics"; see (4). To this end, we used one *Darcy Flow* dataset and *nonlinear diffusion* dataset with training size 30. We varied $\lambda$ from $[0.5, 10^2]$, and run PPI-FNO and PPI-DONet on *Darcy Flow* and *nonlinear diffusion*, respectively. As shown in Fig. 2c and 2d, we can see that across a wide range of $\lambda$ values, PPI-FNO and PPI-DONet can consistently outperform the standard FNO and DONet respectively by a large margin. However, the choice $\lambda$ does have a significant influence on the operator learning performance, and the best choice is often in between. In Appendix Fig. 6c and 6d, we show results on *Ekonal*, which we make similar observations.

**Computational complexity and memory usage.** Our PPI-NO framework conducts alternating updates, and hence needs more training cycles than standard NO. But the time complexity only grows linearly with the number of alternating iterations, rather than quadratically or exponentially. We believe this is reasonable and practically acceptable. For memory usage, Our "pseudo" physics network $\phi$ is very small as compared to the NO component — $\phi$ is simply a pixel-wise FFN coupled with one convolution filter, resulting in a marginal increase in memory cost. Appendix Table 6 shows the parameter count of FNO, DONet and their pseudo-physics-informed version. On average, PPI-FNO increases the number of parameters over FNO by 1.29% while PPI-DONet over DONet by 1.89%.

# 6 Conclusion

We have presented a Pseudo Physics-Informed Neural Operator (PPI-NO) learning framework. PPI-NO is based on our observation that a PDE system is often characterized by a *local* combination of the solution and its derivatives. This characteristic enables the derivation of many training points from the function sampling locations, facilitating learning of the PDE systems through a neural network. While the physics delineated by PPI-NO might not precisely reflect true physical phenomena, our findings reveal that this method significantly enhances the efficiency of operator learning, particularly with limited data quantity.

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

# Appendix

# A  Experimental Details

## A.1  Darcy Flow

We first considered a steady-state 2D Darcy Flow equation (Li et al., 2020c),

$$-\nabla \cdot (a(x)\nabla u(x)) = f(x) \ \ x \in (0,1)^2,$$
$$u(x) = 0 \ \ x \in \partial(0,1)^2, \tag{6}$$

where $u(\mathbf{x})$ is the velocity of the flow, $a(\mathbf{x})$ characterizes the conductivity of the media, and $f(\mathbf{x})$ is the source function that can represent flow sources or sinks within the domain. In the experiment, our goal is to predict the solution $u$ given the external source $f$. To this end, we fixed the conductivity $a$, which is generated by first sampling a Gauss random field $\alpha$ in the domain and then apply a thresholding rule: $a(\mathbf{x}) = 4$ if $\alpha(\mathbf{x}) < 0$, otherwise $a(\mathbf{x}) = 12$. We then used another Gauss random field to generate samples of $f$. We followed (Li et al., 2020c) to solve the PDE using a second-order finite difference solver and collected the source and solution at a $128 \times 128$ grid.

## A.2  Nonlinear Diffusion PDE

We next considered a nonlinear diffusion PDE,

$$\partial_t u(x,t) = 10^{-2}\partial_{xx}u(x,t) + 10^{-2}u^2(x,t) + f(x,t),$$
$$u(-1,t) = u(1,t) = 0, \ \ u(x,0) = 0, \tag{7}$$

where $(x,t) \in [-1,1] \times [0,1]$. Our objective is to predict the solution function $u$ given the source function $f$. We used the solver provided in (Lu et al., 2022), and discretized both the input and output functions at a $128 \times 128$ grid. The source $f$ was sampled from a Gaussian process with an isotropic square exponential (SE) kernel for which the length scale was set to 0.2.

## A.3  Eikonal Equation

Third, we employed the Eikonal equation, widely used in geometric optics and wave modeling. It describes given a wave source, the propagation of wavefront across the given media where the wave speed can vary at different locations. The equation is as follows,

$$|\nabla u(\mathbf{x})| = \frac{1}{f(\mathbf{x})}, \mathbf{x} \in [0,256] \times [0,256] \tag{8}$$

where $u(\mathbf{x})$ is the travel time of the wavefront from the source to location $\mathbf{x}$, $|\cdot|$ denotes the Euclidean norm, and $f(\mathbf{x}) > 0$ is the speed of the wave at $\mathbf{x}$.

In the experiment, we set the wave source at $(0,10)$. The goal is to predict the travel time $u$ given the heterogeneous wave speed $f$. We sampled an instance of $f$ using the expression:

$$f(\mathbf{x}) = \max(g(\mathbf{x}),0) + 1.0,$$

where $g(\cdot)$ is sampled from a Gaussian process using the isotropic SE kernel with length-scale 0.1. We employed the `eikonalfm` library (`https://github.com/kevinganster/eikonalfm/tree/master`) that implements the Fast Marching method Sethian (1999) to compute the solution $u$.

## A.4  Poisson Equation

Fourth, we considered a 2D Poisson Equation,

$$-\Delta u = f, \ \ \text{in } \Omega = [0,1]^2, \ \ u|_{\partial D} = 0. \tag{9}$$

where $\Delta$ is the Laplace operator. The solution is designed to take the form, $u(x_1, x_2) = \frac{1}{\pi K^2}\sum_{i=1}^{K}\sum_{j=1}^{K}a_{ij}(i^2 + j^2)^r \sin(i\pi x_1)\cos(j\pi x_2)$, and $f(x_1, x_2)$ is correspondingly computed via the equation. To generate the dataset, we set $K = 5$ and $r = 0.5$, and independently sampled each element $a_{ij}$ from a uniform distribution on $[0,1]$.

| Parameter count | FNO | PPI-FNO (increase) | DONet | PPI-DONet (increase) |
|---|---|---|---|---|
| Darcy-flow | 1,188,353 | 1,229,476 (+3.46%) | 2,084,704 | 2,125,827 (+1.97%) |
| Nonlinear-diffusion | 1,188,353 | 1,197,220 (+0.75%) | 824,501 | 833,368 (+1.08%) |
| Eikonal | 1,188,353 | 1,197,220 (+0.75%) | 824,501 | 833,368 (+1.08%) |
| Poisson | 1,188,353 | 1,197,220 (+0.75%) | 824,501 | 833,368 (+1.08%) |
| Advection | 1,188,353 | 1,197,220 (+0.75%) | 210,101 | 218,968 (+4.22%) |

Table 6: Parameter counts for FNO and DONet with PPI variations across different problems. The training size is 30.

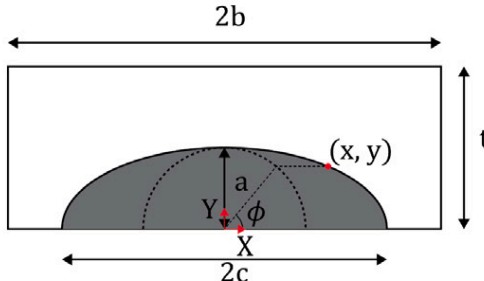

Figure 4: Example of semi-elliptic surface crack on a plates (Merrell et al., 2024).

### A.5 Advection Equation

Fifth, we considered a wave advection equation,

$$\frac{\partial u}{\partial t} + \frac{\partial u}{\partial x} = f, \quad x \in [0,1], \quad t \in [0,1]. \tag{10}$$

The solution is represented by a kernel regressor, $u(\mathbf{x}) = \sum_{j=1}^{M} w_j k(\mathbf{x}, \mathbf{z}_j)$, and the source $f$ is computed via the equation. To collect instances of $(f, u)$, we used the square exponential (SE) kernel with length-scale $0.25$. We randomly sampled the locations $\mathbf{z}_j$ from the domain and the weights $w_j$ from a standard normal distribution.

### A.6 Fatigue Modeling

We considered predicting the SIF values along semi-elliptic surface cracks on plates, as shown in Fig 4. The SIF value can be viewed as a function of the angle $\phi \in [0, \pi]$, which decides the location of each point on the crack surface. The geometry parameters that characterize the crack shape and position were used as the input, including $a/c$, $a/t$ and $c/b$. In the operator learning framework, the input can be viewed as a function with three constant outputs. The dataset was produced via a high-fidelity FE models under Mode I tension (Merrell et al., 2024). Each data instance includes 128 samples of the SIF values drew uniformly across the range of $\phi$.

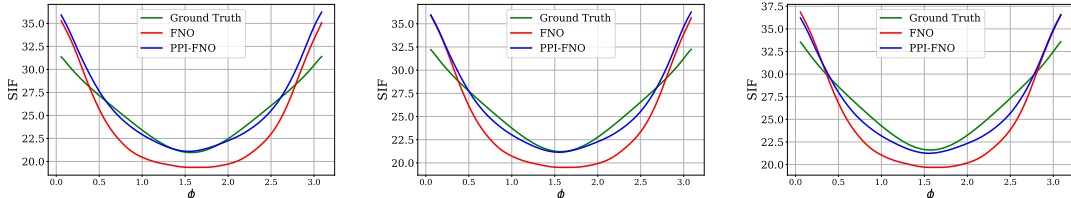

Figure 5: Examples of SIF prediction of FNO and PPI-FNO trained with 600 examples.

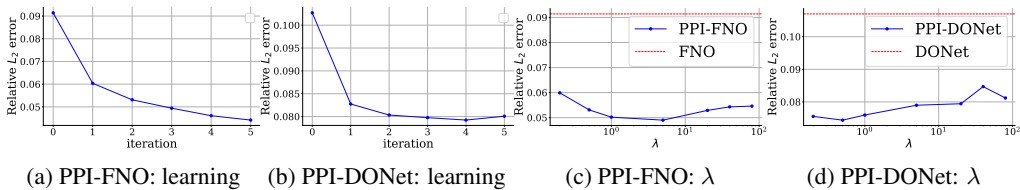

(a) PPI-FNO: learning    (b) PPI-DONet: learning    (c) PPI-FNO: $\lambda$    (d) PPI-DONet: $\lambda$

Figure 6: Learning curve of PPI-FNO (a) and PPI-DONet (b) on *Eikonal* with 30 training examples. Shown in (c) and (d) is how the weight $\lambda$ of "pseudo physics" affects the operator learning performance. The horizontal line in (c) and (d) are the relative $L_2$ error of standard FNO and DONet, respectively.

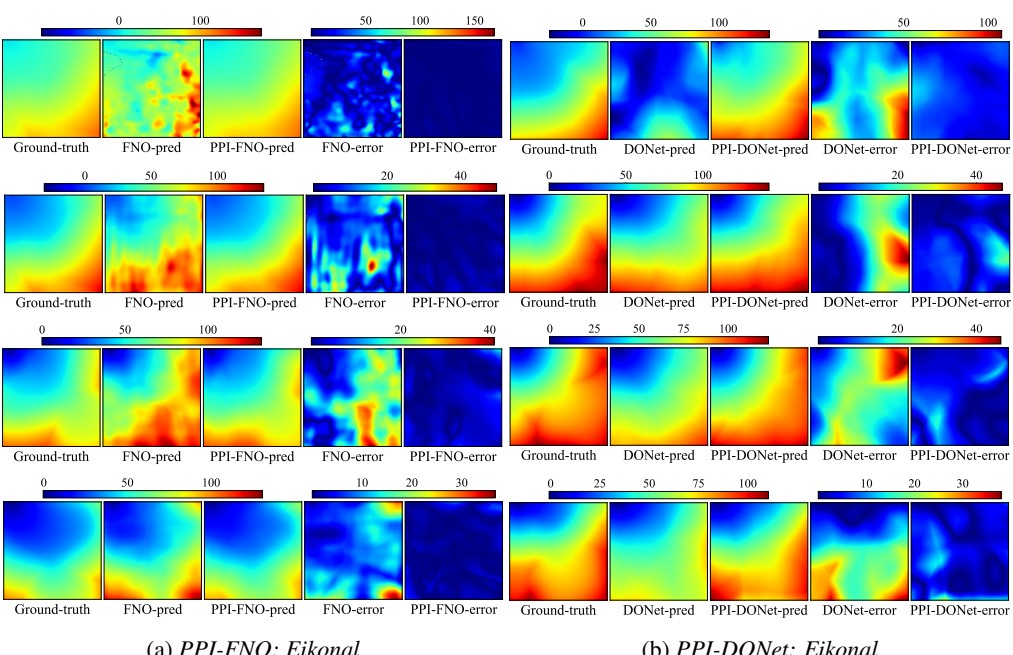

(a) *PPI-FNO: Eikonal*                    (b) *PPI-DONet: Eikonal*

Figure 7: Examples of the prediction and point-wise error of PPI-FNO and PPI-DONet on *Eikonal*. From top to bottom, the models were trained with 5, 10, 20, 30 examples.

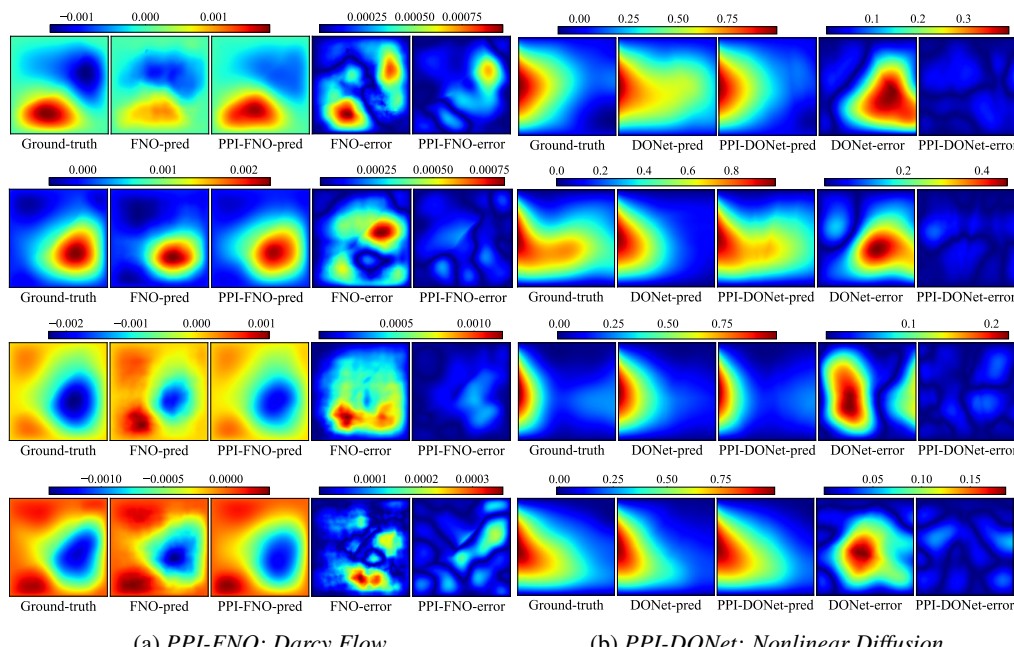

(a) *PPI-FNO: Darcy Flow*  (b) *PPI-DONet: Nonlinear Diffusion*

Figure 8: Examples of the prediction and point-wise error of PPI-FNO and PPI-DONet on *Darcy Flow* and *Nonlinear diffusion*, respectively. From top to bottom, the models were trained with 5, 10, 20, 30 examples.

# B  Limitation and Discussion

Our current method cannot learn PDE representations for which the input function $f$ is the initial condition. In such cases, the mapping from the solution function to the initial condition requires a reversed integration over time, hence we cannot decouple the derivatives. To address this problem, we plan to explicitly model the temporal dependencies in the PDE representation, such as via the neural ODE design (Chen et al., 2018).

