# OpenReview forum: "Pseudo Physics-Informed Neural Operators"
_ICLR.cc/2025/Conference — Submitted to ICLR 2025_

### Official Review · Reviewer_uZpT · 2024-10-24

**Soundness:** 2
**Presentation:** 2
**Contribution:** 2
**Rating:** 3
**Confidence:** 3

**Summary:**

The paper proposes a way to train a solution operator for partial differential equations applicable when only few data points are available. For this, they first apply a system identification technique to approximate the underlying partial differential equation and then use this equation for physics-informed training of their solution operator. They evaluate their method on 5 non-trivial partial differential equations.

System identification is a well-studied field and physics-informed training of neural operators has been done before, as the authors correctly describe. The novelty lies in their combination to train a solution operator with only few data points. To my knowledge, this has indeed not been done before. The idea is simple and makes sense.

I think the biggest weakness is that the straight-forward way to train a solution operator with few data is not discussed and therefore, also not part of their experiments: After the PDE is approximated, one could use this to generate new data and train the solution operator in a supervised setting with many data points. I would expect this to work better than the author’s method since neural networks are easier to optimize with data sets than with a physics-informed loss. Their only baseline is supervised training with few data points. As I expect training on few data points to require much less time than physics-informed training, I don't consider this a fair comparison. A convincing benchmark would require reporting the actual time spent on training and then further spent the same amount of time on a decent baseline, for instance, the one outlined above. For example, half of the time for the generation of further solution-source terms, and the other half for training the solution operator.

Another issue is that the author's method is basically using two other techniques from two different subfields in sequence rather than coupling the underlying principles into an improved method. While this is not inherently negative, it raises the expectation for a more generalized argumentation. For instance, including more than one specific technique for each of the subfields, such as another technique for system identification. Or making a theoretical argument about how the individual errors of the PDE approximation in the first step and the solution operator approximation add up to the total error.

Some minor points:
- I think many details on the experiments (convolutional kernel size, activation functions, frameworks used,...) should be moved to the appendix
- Equation 2 indicates that the first neural network (denoted by phi in the paper) acts on quantities at a specific spatial point while Figure 1 indicates phi acts on entire fields on the spatial domain. This should be clarified.
- The related work section should mention some works on system identification.

**Strengths:**

See above.

**Weaknesses:**

See above.

**Questions:**

- The abstract states that the method ‘enhances the accuracy of current operator learning
models, particularly in data scarce scenarios’.  As I understand the paper, the presented method is  suited only for data-scarce scenarios.
- Why are physics-informed losses introduced as a regularization technique? (in the introduction)

---

> ### Author Response · Authors · 2024-11-24
>
> We thank the reviewer for the comments. C: comments; R: response
>
> >C1: I think the biggest weakness is that the straight-forward way to train a solution operator with few data is not discussed and therefore, also not part of their experiments: After the PDE is approximated, one could use this to generate new data and train the solution operator in a supervised setting with many data points. I would expect this to work better than the author’s method since neural networks are easier to optimize with data sets than with a physics-informed loss. Their only baseline is supervised training with few data points. As I expect training on few data points to require much less time than physics-informed training, I don't consider this a fair comparison. A convincing benchmark would require reporting the actual time spent on training and then further spent the same amount of time on a decent baseline, for instance, the one outlined above. For example, half of the time for the generation of further solution-source terms, and the other half for training the solution operator.
>
> R1: we respectfully disagree with the assessment that generating new data for supervised training would outperform our method. In fact, **we conducted experiments along these lines early in our work and found that this approach yielded inferior results compared to our proposed regularization framework**. Below are the experimental results on the Darcy dataset under comparable settings:
>
> Training of operator $f\rightarrow u$ with 200 generate samples, test error:
> |  Relative $L_2$ | Training size=5  |  10 | 20  |  30 |
> |---|---|---|---|---|
> | train generated data with original data| 0.4492(0.0196) | 0.2638(0.0132) | 0.1646(0.0168) | 0.1013(0.0023)|
> |  Ours    | 0.1716(0.0048)    | 0.0956(0.0084)   | 0.0680(0.0031)    | 0.0642(0.0010)  |
>
> As shown, directly adding virtual data points resulted in higher errors compared to our method, particularly in low-data scenarios. One potential reason for this is that our regularization approach allows us to tune the strength of the pseudo-physics term, effectively distinguishing between virtual data points and real data points during training. This adaptability contributes to the superior performance of our framework.
>
>
> >C2: The abstract states that the method 'enhances the accuracy of current operator learning models, particularly in data scarce scenarios'. As I understand the paper, the presented method is suited only for data-scarce scenarios.
>
> R2: You are correct that our method is particularly suited for data-scarce scenarios. In realistic settings, it is often impractical or impossible to obtain detailed knowledge of the governing physical laws, and data is frequently limited due to the high cost of simulations or the difficulty of acquiring measurement data.
>
> The data-scarcity issue is **not uncommon**, especially in practical domains where simulations are computationally expensive, or experimental data collection is resource-intensive. Our method is specifically designed to address this significant challenge, providing a solution for operator learning under these constraints.
>
> >C3: Why are physics-informed losses introduced as a regularization technique? (in the introduction)
>
> R3: Great question,  Physics-informed losses are introduced as a regularization technique to leverage the pseudo-physics learned through our inverse model $\phi$. The goal is for the pseudo-physics to guide and regularize the training process of the operator model, improving its performance when working with limited data. This approach helps bridge the gap between sparse data availability and the need for accurate operator learning by enforcing consistency with the underlying physics.

---

> > ### Comment · Area_Chair_c3AU · 2024-11-26
> >
> > Dear reviewer,
> >
> > Please make to sure to read, at least acknowledge, and possibly further discuss the authors' responses to your comments. Update or maintain your score as you see fit.
> >
> > The AC.

---

> > > ### Comment · Reviewer_uZpT · 2024-11-27
> > >
> > > Dear Authors,
> > >
> > > Thank you for your response.
> > >
> > > In my opinion, your findings are somewhat counterintuitive due to the optimization difficulties associated with PINNs, as well as PINOs. Therefore, an accurate experimental evaluation seems crucial to me here. The table in R1 of your answer is a step in the right direction. If I connected this correctly to the results in the paper, supervised training with 200 generated data points is better than without any generated data. I would also like to repeat my recommendation to report the actual training time (and potentially the learning curves) when comparing the different methods. I suspect that the PINO variant will take significantly longer to train than the variant with 200 generated data, but this is just an assumption and I could be mistaken.

---

### Official Review · Reviewer_F2kC · 2024-10-28

**Soundness:** 3
**Presentation:** 4
**Contribution:** 3
**Rating:** 6
**Confidence:** 4

**Summary:**

The paper adapts physics-informed neural operators to settings without knowledge of the underlying PDE by approximating while learning the neural operator. This is then shown to improve the performance of neural operators in the case of scarce data.

**Strengths:**

- The approach is interesting, novel, and well-reasoned.
- The ablation studies for different components are appreciated.
- Thorough evaluations show the effectiveness of the method.
- Clear presentation, easy to follow

**Weaknesses:**

- The presented approach can only learn the operator mapping the source function to the solution function. That this is not the standard operator learning problem setting (as for example discussed in the FNO paper) is only mentioned in the limitations section at the end of the appendix. It would be helpful to mention the focus during the problem formulation already and put the limitation section in the main paper.
- The training set sizes seem random and sometimes do not cover a broad range. This is the most significant for the SIF dataset. It would have been interesting to see different experiments with dataset sizes covering a broader range like 10, 100, 1000
- Timing: It would be interesting to have an actual time comparison between the methods. Furthermore, since the idea is that this idea saves time as less data has to be generated, a comparison to the time it takes to compute more data would be interesting.

Minor Notes:
- You often write feedforward layer/network, when I think you mean fully-connected layer/network. A convolutional layer is, for example, also a feedforward layer.
- p.4: You state that you use "numerical difference" to compute the derivatives. Do you mean finite differences?
- Eq. 4: p(f) is not defined
- Figure 2 seems to be too early
- p. 10, l.521: "the best choice \[of $\lambda$\] is often in between" In between what? This seems to be a very vague statement

**Questions:**

- The experiments in Table 3 and and Table 4a seem very similar. Can you explain the main difference (apart from including FNO) is?
- While I understand that it is not the idea to use PPI-FNO when the PDE is known, it would be interesting to see a comparison between PPI-NO und PI-NO, to learn about the loss by approximating the PDE instead of using the correct one. Can you run some experiments with PI-NO for a comparison?

---

> ### Author Response · Authors · 2024-11-24
>
> We thank the reviewer for the comments. C: comments; R: response
>
> >C1:p.4: You state that you use "numerical difference" to compute the derivatives. Do you mean finite differences?
>
> R1: Yes, they are essentially the same in this context.
>
> >C2: The experiments in Table 3 and and Table 4a seem very similar. Can you explain the main difference (apart from including FNO) is?
>
> R2: Thank you for your observation. The main difference between the experiments in Table 3 and Table 4a lies in the inclusion of derivative information in the model inputs:
>
> Table 3: The FFN model incorporates derivative information in its inputs, which is key to its performance.
>
> Table 4a: In our ablation study, the baseline models (MLP and FNO) do not include derivative information in their inputs.
>
> We also noticed a typo on Line 472 in the manuscript when describing the model input. It should read as "not included," and we will correct this in the revised version.
>
> >C3: While I understand that it is not the idea to use PPI-FNO when the PDE is known, it would be interesting to see a comparison between PPI-NO und PI-NO, to learn about the loss by approximating the PDE instead of using the correct one. Can you run some experiments with PI-NO for a comparison?
>
> R3: We appreciate the reviewer's suggestion. we agree that comparing with a physics-informed neural operator (PI-NO) using correct physics is an interesting avenue. Below, we present the results of such a comparison on the Poisson data:
>
> |  Relative $L_2$ | Training size=5  |  10 | 20  |  30 |
> |---|---|---|---|---|
> | PI-NO with 200 generated samples| 0.1890(0.0042) | 0.0863(0.0020) | 0.0596(0.0168) | 0.0492(0.0003)|
> |  Ours    |  0.1437(0.0062)    | 0.0771(0.0018)   |  0.0544(0.0009)    |  0.0458(0.0003)  |
>
> As the training size increases, the relative $L_2$ errors of the two methods become very close. This convergence is likely due to two factors: (1) our alternative training framework inherently involves running more epochs in total compared to PI-NO, although in this case, we capped PI-NO training at 1000 epochs per sample for computational efficiency; and (2) our alternative training framework demonstrates clear performance gains when the training size is small (e.g., size = 5).

---

> > ### Comment · Area_Chair_c3AU · 2024-11-26
> >
> > Dear reviewer,
> >
> > Please make to sure to read, at least acknowledge, and possibly further discuss the authors' responses to your comments. Update or maintain your score as you see fit.
> >
> > The AC.

---

> ### Comment · Reviewer_F2kC · 2024-11-26
>
> Thank you to the authors for their reply. I have some remaining and follow up questions:
>
> __Remaining__
>
> Please address also the weaknesses mentioned in the initial review, you have ignored them completely.
>
>
> __Follow up__
>
> I have never heard the term "numerical differences" before and a google search does not yield any results either. I recommend changing to "finite differences".
>
> To understand Table 4 correctly: FNO, MLP, and "ours" do all not include derivative information? Why did you choose this setting?
>
> Why is PI-NO worse than your method? That was not to be expected, since your method is only approximating the PDE? Especially in this low data setting it should be the other way around as the estimate of the PDE should be worse as well.

---

### Official Review · Reviewer_VSib · 2024-10-28

**Soundness:** 3
**Presentation:** 3
**Contribution:** 3
**Rating:** 5
**Confidence:** 4

**Summary:**

In complex systems with minimal information of the underlying physics, it is difficult to model physics based losses. To overcome this issue, the authors propose a surrogate model that learns the inverse mapping between the solution at discrete points, its derivatives and the source term at the corresponding discrete points. This model effectively serves as the “teacher model” for a neural operator framework that learns the solution from the source term. The derivatives of the solution are computed based on numerical differences. It is pseudo physics informed because the operator is trained using the surrogate model rather than loss functions and residuals defined over the actual PDE.

**Strengths:**

Originality: The use of the inverse model as the ground truth in cases where the data is sparse and the governing PDE is unknown is quite promising because in a lot of applied settings, it is not always the case that a governing PDE is known.

Quality: The intuition of the paper is quite clear. There are thorough experiments on standard benchmark datasets. The authors perform several ablations to substantiate their claims. The figures clearly indicate the message the authors are trying to convey.

Significance: This is a novel idea that builds upon the Physics-informed ML literature, combining inverse-PDE estimates into the learning pipeline as an alternative to physics based residuals and losses.

**Weaknesses:**

They use the neighborhood information captured within the convolution layers as a way to compensate for errors in numerical differences. A graph neural operator would be both discretization agnostic and would be better for capturing neighborhood information.

The training dataset seems quite low. It is not clear whether 5 examples indicate 5 instances of the same PDE with different co-efficients or whether it’s 5 different sparse representations, with the same co-efficients.

The property of neural operator is that it’s discretization agnostic. The authors don’t mention what discretizations they tested. 128x128 grid is not indicative of the discretization, but rather the resolution. By this I mean that this setting could be a set of densely located 128x128 points in a very small area within a large mesh or a set of 128x128 sparse points spread over the entire mesh.

While comparing against data driven FNO models is a good baseline, the authors propose this architecture as a substitute for Physics informed ML. Therefore, it would be appropriate to show how this scales against PINNs and PINOs.

In the FNO paper, the models were trained on training sets with 1000 instances. However, the authors here use a significantly smaller training dataset. Could it be possible that the failure scenarios shown in Figure 3. are because the FNO models require a larger training set to converge? Perhaps a more fair comparison would be to train both the FNO model and the PPI-FNO model on the larger dataset.
It seems unreasonable to think that a system is so sparse that the training dataset only has 5 instances. Moreover, it is not clear whether sparsity refers to the size of the training dataset or the number of points within the mesh (sparse discretization).

**Questions:**

1.	When the source terms and the boundary conditions are known, the PDEs can be estimated using Monte-Carlo Walk-on-Spheres (WOS). Neural Walk-on-spheres trains neural networks based on WOS estimates. How does the error rate of the Surrogate model compare against random-walks that accumulate the source term over the green function?

2.	What is the justification for using convolution neural networks as the surrogate model, to capture neighborhood information? A radius based graph neural network is discretization agnostic and works especially well in sparse settings.

3.	The surrogate model is not discretization agnostic. The functions sampled would have to be the same discretization as it was trained on. Which would mean that the neural operator model can predict any sparse distribution of points, but the second loss term (i.e. the surrogate model) has to be a fixed discretization. This seems like a bottleneck. Were there reasons for not making the second model a neural operator. Perhaps using [1] would be a good way to ensure operator learning through the entire pipeline.

[1] Wang, Tian, and Chuang Wang. "Latent Neural Operator for Solving Forward and Inverse PDE Problems." arXiv preprint arXiv:2406.03923 (2024).

---

> ### Author Response · Authors · 2024-11-24
>
> We thank the reviewer for the comments. C: comments; R: response
>
> >C1:The training dataset seems quite low. It is not clear whether 5 examples indicate 5 instances of the same PDE with different co-efficients or whether it's 5 different sparse representations, with the same co-efficients.
>
> R1: Thank you for pointing this out. To clarify, we follow the standard neural operator (NO) testing settings, where we use the same PDE for all examples. However, each instance is associated with a different source/input function, leading to completely different solution functions. The objective is to learn the operator corresponding to the same PDE, capturing its general behavior across these diverse input-output pairs.
>
>
> >C2:The property of neural operator is that it's discretization agnostic. The authors don't mention what discretizations they tested. 128x128 grid is not indicative of the discretization, but rather the resolution. By this I mean that this setting could be a set of densely located 128x128 points in a very small area within a large mesh or a set of 128x128 sparse points spread over the entire mesh.
>
> R2: To clarify, we use a regularly spaced grid for our experiments, which is essential for enabling the operation of the Fourier Neural Operator (FNO). Specifically, the 128x128 grid represents a uniform resolution across the domain, rather than a set of densely or sparsely located points over a varying area or mesh.
>
>
> >C3:In the FNO paper, the models were trained on training sets with 1000 instances. However, the authors here use a significantly smaller training dataset. Could it be possible that the failure scenarios shown in Figure 3. are because the FNO models require a larger training set to converge? Perhaps a more fair comparison would be to train both the FNO model and the PPI-FNO model on the larger dataset. It seems unreasonable to think that a system is so sparse that the training dataset only has 5 instances. Moreover, it is not clear whether sparsity refers to the size of the training dataset or the number of points within the mesh (sparse discretization).
>
> R3: Thank you for your comment. The primary motivation of our work is to address **data sparsity**, specifically in scenarios where the number of training instances is small. In many real-world applications, such as scientific computing and engineering simulations, obtaining large datasets of high-fidelity solutions can be prohibitively expensive or infeasible. Our framework is designed to improve performance under such constraints.
>
> To clarify, in this context, data sparsity refers to the number of training instances, not the sparsity of points within the mesh (sparse discretization). For instance, in the SIF dataset, we used 500-600 examples, which is already **significantly fewer** than the 1000+ instances typically used in FNO setups. This dataset was chosen to balance the trade-off between realistic data availability and sufficient diversity for model training.
>
> >C4: What is the justification for using convolution neural networks as the surrogate model, to capture neighborhood information? A radius based graph neural network is discretization agnostic and works especially well in sparse settings.
>
> R4: Great question. In our  $\phi$ network, we chose convolutional neural networks (CNNs) because, when the discretized versions of u and f are represented as grid-contained points, the neighborhood information helps to reveal hidden relationships, particularly those related to derivatives between neighboring points. By incorporating a convolutional layer, our network can effectively learn these hidden trends at each point, which contributes to capturing the underlying physics. Our ablation study in Table 4 further demonstrates the significant improvement introduced by the convolutional layer.
>
> We agree that graph neural networks (GNNs) could offer an interesting alternative, especially in discretization-agnostic or sparse settings. However, our current architecture-a combination of a single convolutional layer with a pointwise MLP-has already shown strong performance in discovering hidden physics and improving operator learning. We are open to exploring alternative architectures, including GNNs, in future work to further enhance our framework.
>
> >C5: The surrogate model is not discretization agnostic. The functions sampled would have to be the same discretization as it was trained on. Which would mean that the neural operator model can predict any sparse distribution of points, but the second loss term (i.e. the surrogate model) has to be a fixed discretization. This seems like a bottleneck. Were there reasons for not making the second model a neural operator. Perhaps using [1] would be a good way to ensure operator learning through the entire pipeline.
>
> R5: Thanks for the great suggestion. We will try them in future experiments.

---

> ### Comment · Reviewer_VSib · 2024-11-24
>
> R1, R2: Thank you for providing clarifications for these questions.
>
> R3: While the results shown in this paper show the approach outperforming baseline FNO models on sparse settings, it's not clear to me whether the FNO is overfitting. I am unable to rule out the following reasons why the FNO model is not performing as well as the proposed model:
>  1. In Tables 1 and 2, in a lot of the examples shown, the loss is of the same order of magnitude. (i.e. for eg.  e^-1 on SIF). The improvement in performance could just be a fluctuation due to a difference in parameter count between the two models. Maybe providing the training trends between the two models will help understand if the FNO converged fully.
>
> 2. "FOURIER NEURAL OPERATOR FOR PARAMETRIC PARTIAL DIFFERENTIAL EQUATIONS", [Li, 2020] provide the optimal settings to achieve the best performance with vanilla-FNO. Assuming that the training data was not sparse, would the proposed approach still outperform FNO? If not, then it would be helpful to provide a graph (plot) of the degree of reduction in training-dataset size vs loss to understand the threshold for training dataset size that would lead to the proposed model outperforming vanilla FNO.
>
> A followup question is why the authors choose to compare against vanilla FNO and deep-O-Net when several papers have been published since then with better results. For example, "Solving Poisson Equations using Neural Walk-on-Spheres", [Nam, 2024], while not directly relevant to the authors' work, achieved a loss of the order of e-3 on Poisson equation while the proposed model achieved a loss of e^-1. Similarly, "Gnot: A general neural operator transformer for operator learning", achieved a loss of 1e-2 on Darcy Flow.
> Perhaps the authors could provide an application where a loss of e-1 is acceptable.
>
> R4, R5: This doesn't seem convincing. The justification for using a Fourier Neural Operator layer in the main model goes against not using the same in the pseudo physics network.

---

> > ### Author Response · Authors · 2024-11-24
> >
> > >C1: While the results shown in this paper show the approach outperforming baseline FNO models on sparse settings, it's not clear to me whether the FNO is overfitting. I am unable to rule out the following reasons why the FNO model is not performing as well as the proposed model: In Tables 1 and 2, in a lot of the examples shown, the loss is of the same order of magnitude. (i.e. for eg. e^-1 on SIF). The improvement in performance could just be a fluctuation due to a difference in parameter count between the two models. Maybe providing the training trends between the two models will help understand if the FNO converged fully.
> >
> > R1: In all our experiments, the baseline FNO was trained for a maximum of 150 epochs. From our observations, the FNO typically converges around 120 epochs, with minor variations depending on the specific dataset. We ensured that FNO was fully trained to convergence to provide a fair comparison.
> >
> > We include the training trends for Darcy data and Poisson data in the following link. These figures will provide a clearer comparison and confirm that the FNO was fully trained in our experiments.
> >
> > [darcy_fno_test_loss_trend] https://www.dropbox.com/scl/fi/egdms0e6bu5zc1jcxfe17/darcy_fno_test_loss_trend.png?rlkey=z38vo31z0tzd2t3qht5gzuz61&st=njmysbv5&dl=0
> >
> > [poisson_fno_test_loss_trend] https://www.dropbox.com/scl/fi/53rts1l8makkzq04yj58p/poisson_fno_test_loss_trend.png?rlkey=nxuqfmw7qgtdndodrt56s2mp8&st=ld5po52n&dl=0
> >
> > >C2:"FOURIER NEURAL OPERATOR FOR PARAMETRIC PARTIAL DIFFERENTIAL EQUATIONS", [Li, 2020] provide the optimal settings to achieve the best performance with vanilla-FNO. Assuming that the training data was not sparse, would the proposed approach still outperform FNO? If not, then it would be helpful to provide a graph (plot) of the degree of reduction in training-dataset size vs loss to understand the threshold for training dataset size that would lead to the proposed model outperforming vanilla FNO.
> >
> > R2: We believe that this expectation goes beyond the scope of our paper. Our work is specifically designed to address the sparse data learning problem, which we consider highly relevant to real-world scenarios where acquiring large datasets is impractical or costly. The goal of our approach is not to outperform FNO in settings with abundant training data but to provide a robust solution in data-scarce conditions.
> >
> > It is important to note that no single method can guarantee optimal performance across all scenarios. Requiring a model to perform best in both sparse and abundant data settings is neither realistic nor reasonable, as each approach is tailored to address specific challenges.
> >
> > >C3: R4, R5: This doesn't seem convincing. The justification for using a Fourier Neural Operator layer in the main model goes against not using the same in the pseudo physics network.
> >
> > R3: We did experiment with using a Fourier Neural Operator (FNO) layer in our pseudo-physics network early in our work. However, we found that it performed worse than our MLP with convolutional layers. Below are the results from an experiment using the Darcy dataset:
> >
> > Error of predicting $f$:
> > |  Relative $L_2$ | Training size=5  |  10 | 20  |  30 |
> > |---|---|---|---|---|
> > | FNO with derivatives |  0.7182(0.0349)   | 0.5807(0.0116)   |   0.4169(0.0120)  | 0.3325(0.0092)   |
> > |  Ours |  0.2285(0.0147)   | 0.1392(0.0080)   |   0.0898(0.0046)  | 0.0688(0.0032) |
> >
> > The performance of the FNO with derivatives in the pseudo-physics network was very similar to the result in our ablation study **(Table 4a)**, where we used FNO without derivative information. This demonstrates that our pseudo-physics network with MLP and convolutional layers outperforms the FNO-based approach in this context.

---

> > > ### Comment · Reviewer_VSib · 2024-11-25
> > >
> > > R1. This experiment shows that in the sparse setting, FNO converges at a higher error than the specified model
> > >
> > > R2. This further makes me skeptical about whether the model is overfitting. If the authors claim that sparse training data is sufficient to generalize to validation dataset, then on a larger training dataset, the performance should be comparable or better. If the model fails to outperform FNO when the training dataset is larger, it could indicate issues with generalization.
> > >
> > > Based on the responses provided by the authors, I'm not convinced that the experiments performed substantiate the theoretical claims. While I find the paper interesting and the authors have conducted thorough experiments, I think the experiments may have used inadequate settings. Furthermore, I find the authors' answers to questions by the other reviewers unconvincing. I will not change my score.
> > >
> > > I would like to provide actionable feedback regarding the experiments section:
> > >
> > > 1. Prove that the model outperforms FNO, GNOT, DINo, IPOT etc. in all reasonable settings - This backs the theoretical claims in the earlier sections
> > > 2. Prove that the model generalizes well to sparse training data - This provides evidence of improved efficiency compared to SOTA models. (The authors have already attempted to do this in the paper, but it would help to include newer SOTA models)

---

### Official Review · Reviewer_T4BB · 2024-11-02

**Soundness:** 2
**Presentation:** 2
**Contribution:** 2
**Rating:** 3
**Confidence:** 4

**Summary:**

The authors propose the Pseudo Physics-Informed Neural Operator (PPI-NO), which couples the existing concepts of physics discovery and neural operator learning. In particular, a surrogate partial differential equation (PDE) representation is learned from data using a neural network. Afterwards, the neural-network-PDE model is used as a regularizer to refine the training of the neural operator. The authors claim that the coupling helps the neural operator learn effectively in the low data limit.

**Strengths:**

The paper is written clearly and has appropriate results to support the authors' claim. Further, the paper proposes the integration of the non-trivial concepts of physics discovery and neural operator learning, which is an important problem.

**Weaknesses:**

1.The idea of coupling physics discovery with NN has been explored earlier. For e.g., see PINN-SR [1].

2. The basic idea of the manuscript is problematic. The discovered "pseudo" physics is not exact and hence is of much lower-fidelity (and is unlikely to generalize). The data available is of higher fidelity. Therefore a composite loss function where one term is of higher fidelity and the other is of lower fidelity will, in theory, stop the model from generalization. This fact has been previously pointed on in [2] and as a remedy transfer learning was proposed.

3. Even by incorporating rudimentary physics information, a significant decrease in error is not observed in Table 1 (which is not totally unexpected given the point above). In the results of the DONet-Darcy flow, DONet-Diffusion, and all Poisson and advection equations, the reduction in error is minimal, which makes the contribution of the discovered physics marginal.

4. Like any other basis function-based physics-discovery algorithms, this framework also requires careful selection of the derivatives, which limits the proposed framework's applicability. It is evident in Table 1. Even when the training data is increased, the relative error increases instead of decreasing in some cases. This may be due to faulty physics identification. I will also add that since the exact terms are not known, using a L2 loss in generally not preferred (as with L2 error, even those terms that are supposed to be absent will have non-zero weights). This contributes to the error in equation discovery and hence, the accuracy of the overall method.

5. Important aspects like the effect of incorporating physics on the zero-shot prediction on super- and sub-resolutions, as well as generalization to out-of-distribution input, have not been studied. These are required to gauge the strength of the proposed framework correctly.

[1] Chen, Zhao, Yang Liu, and Hao Sun. "Physics-informed learning of governing equations from scarce data." Nature communications 12.1 (2021): 6136.
[2] Chakraborty S. Transfer learning based multi-fidelity physics informed deep neural network. Journal of Computational Physics. 2021 Feb 1;426:109942.

**Questions:**

1. l083. You define f(x) as the source function. I believe neural operators go beyond simply source functions to solution mapping.
2. l087. \mathbb{F} and \mathbb{U} are not defined.
3. l147. How order of derivatives should be chosen?
4. Eq. (5). Why generate N' samples in the second term? Instead, why can we not use the available N samples from the first term?
5. In section 4, important literature in this area are missing. For example, SNO [1], CNO [2], LNO [3], and PIWNO [4] are not reviewed.
6. l301. Why are the same derivatives not used across all the examples? How are they chosen?
7. l302. For the SIF example, why are polynomials of the derivatives not used?
8. l311. What do the iterations denote?
9. l317. For the SIF example, 400-600 training samples are used. Obtaining such a training set using high-fidelity crack simulations is very costly. This completely defeats the purpose of the proposed framework.
10. Table 1. Why does the error in DONet-Darcy, DONet-Poisson, and DONet-Advection examples increase with the increase in training data?
11. In Table 2. The decrease in error in the case of PPI-NO is very marginal. This indicates that the incorporation of rudimentary physics is ineffective in complex problems like the SIF prediction.
12. l413. Should the baseline comparison be moved to an ablation study in the given setup? Otherwise, the comparison for physics accuracy should be made with dedicated physics discovery algorithms like PINN-SR [5].


[1] Fanaskov, Vladimir Sergeevich, and Ivan V. Oseledets. "Spectral neural operators." Doklady Mathematics. Vol. 108. No. Suppl 2. Moscow: Pleiades Publishing, 2023.

[2] Raonic, Bogdan, et al. "Convolutional neural operators for robust and accurate learning of PDEs." Advances in Neural Information Processing Systems 36 (2024).

[3] Cao, Qianying, Somdatta Goswami, and George Em Karniadakis. "Laplace neural operator for solving differential equations." Nature Machine Intelligence 6.6 (2024): 631-640.

[4] Navaneeth, N., Tapas Tripura, and Souvik Chakraborty. "Physics informed WNO." Computer Methods in Applied Mechanics and Engineering 418 (2024): 116546.

[5] Chen, Zhao, Yang Liu, and Hao Sun. "Physics-informed learning of governing equations from scarce data." Nature communications 12.1 (2021): 6136.

---

> ### Author Response · Authors · 2024-11-24
>
> We thank the reviewer for the comments. C: comments; R: response
>
> >C1:The basic idea of the manuscript is problematic. The discovered "pseudo" physics is not exact and hence is of much lower-fidelity (and is unlikely to generalize). The data available is of higher fidelity. Therefore a composite loss function where one term is of higher fidelity and the other is of lower fidelity will, in theory, stop the model from generalization. This fact has been previously pointed on in [2] and as a remedy transfer learning was proposed.
>
> R1: we believe there is a logical inconsistency in the argument. While it is true that high-fidelity data provides more precise information, the **issue** in our case is **the limited quantity of such data, which makes it insufficient for training a reliable high-fidelity model on its own.**
>
> In this scenario, it is not evident how incorporating low-fidelity equations would "ruin the training." On the contrary, **robust low-fidelity equations can serve as a regularizing influence, guiding the model to generalize better in data-scarce conditions**. This is especially critical when direct reliance on sparse high-fidelity data may lead to overfitting.
>
> >C2: l083. You define f(x) as the source function. I believe neural operators go beyond simply source functions to solution mapping.
>
> R2: Yes, in many practical scenarios, both f(x) (the source function) and u(x) (the solution) are often unknown or partially observed. Our proposed framework is specifically designed to tackle these more challenging problems by improving the performance of neural operators under such conditions.
>
> >C3: l087. \mathbb{F} and \mathbb{U} are not defined.
>
> R3: \mathbb{F} and \mathbb{U} are two function spaces (e.g., Banach spaces) and U is the solution of F.
>
> >C4: l147. How order of derivatives should be chosen?
>
> R4: from L477, we did ablation study about to chose order of derivative. After compare choices of different derivatives, we chose the order of derivatives up to 2.
>
> >C5: Eq. (5). Why generate N' samples in the second term? Instead, why can we not use the available N samples from the first term?
>
> R5: Great question. Our method is designed to address scenarios with **limited data**, which is a common challenge in real-world operator learning. In such cases, N is already small, and relying solely on these N samples would limit the diversity of input functions, which is critical for learning a robust operator.
>
> To address this, we generate N'additional samples to cover a wider range of input functions. This allows the learned operator to better satisfy the discovered pseudo-physics and generalize effectively. The generated samples act as an augmentation strategy, reinforcing the learning process in data-scarce conditions.
>
> >C6: In section 4, important literature in this area are missing. For example, SNO [1], CNO [2], LNO [3], and PIWNO [4] are not reviewed.
>
> R6: Thank you for your suggestion, we will cite and discuss them in our manuscript.
>
> >C7: l301. Why are the same derivatives not used across all the examples? How are they chosen?
>
> R7: In general, the order of derivatives remains consistent across examples. However, for cases such as Darcy Flow, Eikonal, and Poisson equations, the solutions depend **only on spatial variables** (x1 and x2) and **not on time** (t). In contrast, other examples involve dependencies on **both spatial (x) and temporal (t) variables**.
>
> To ensure clarity and better understanding, we explicitly specify the derivatives for each case based on the known equations. This approach highlights the unique dependencies of each problem while maintaining consistency in the methodology.
>
> >C8: l302. For the SIF example, why are polynomials of the derivatives not used?
>
> R8: Our method is designed to be general and does not impose a predefined strong form of the PDE. Instead, we aim to learn the governing equations directly from the data, allowing the model to adapt to the underlying physics without restrictive assumptions.
>
> Neural networks are sufficiently expressive to approximate polynomials if the true PDE involves polynomial terms. By not explicitly enforcing a polynomial structure, we ensure that our approach remains flexible and applicable to a broader range of problems.
>
> >C9: l311. What do the iterations denote?
>
> R9: As show in figure 6, the alternative training iterations will help to continue improve the performance until they converge.

---

> ### Author Response · Authors · 2024-11-24
>
> >C10: l317. For the SIF example, 400-600 training samples are used. Obtaining such a training set using high-fidelity crack simulations is very costly. This completely defeats the purpose of the proposed framework.
>
> R10:We agree that obtaining 400-600 high-fidelity samples can be costly. However, this is significantly fewer than the number of samples required by traditional methods, which can demand 10 times more examples to achieve comparable accuracy.
>
> It is important to note that while our framework aims to reduce the reliance on extensive high-fidelity data, expecting it to perform optimally with just a handful of samples (e.g., 5) is neither realistic nor aligned with the challenges of complex scenarios like crack simulations. Our approach strikes a balance by minimizing data requirements while maintaining strong performance, and we believe this represents a meaningful step toward solving these challenging problems.
>
>
> >C11: Table 1. Why does the error in DONet-Darcy, DONet-Poisson, and DONet-Advection examples increase with the increase in training data?
>
> R11: We believe there might be a misunderstanding of the data presented in Table 1. In the cases of DONet-Darcy, DONet-Poisson, and DONet-Advection, the error **decreases** as the amount of training data increases. Additionally, the rate of error reduction improves with more training data, highlighting the effectiveness of our framework.
>
> >C12: In Table 2. The decrease in error in the case of PPI-NO is very marginal. This indicates that the incorporation of rudimentary physics is ineffective in complex problems like the SIF prediction.
>
> R12: we believe the observation that "the improvement is marginal" is **not accurate**. As shown in Table 2, the minimal decrease in error for PPI-NO is 18%, and in approximately 70% of the cases, the error reduction exceeds 30%. These results indicate substantial improvements rather than marginal ones.
>
> As mentioned earlier, SIF prediction represents a significantly more complex and realistic problem compared to single PDE scenarios, which naturally makes achieving large improvements more challenging. Despite this, the performance gains we observe are still meaningful, demonstrating the utility of our approach even in these difficult cases. On the contrary, imagine without our method, a much greater number of examples would be required to train an equally well NO, which will present much higher data cost.
>
> >C13: l413. Should the baseline comparison be moved to an ablation study in the given setup? Otherwise, the comparison for physics accuracy should be made with dedicated physics discovery algorithms like PINN-SR [5].
>
> R13: Thank you for your suggestion, we will consider to move this to l468 and conclude the ablation study together.

---

> > ### Comment · Area_Chair_c3AU · 2024-11-26
> >
> > Dear reviewer,
> >
> > Please make to sure to read, at least acknowledge, and possibly further discuss the authors' responses to your comments. Update or maintain your score as you see fit.
> >
> > The AC.

---

### Official Review · Reviewer_KpCq · 2024-11-03

**Soundness:** 4
**Presentation:** 3
**Contribution:** 4
**Rating:** 6
**Confidence:** 4

**Summary:**

Paper introduces pseudo physics-informed Neural operators tailored for complex scenarios where physics is not fully known and data is sparse. It presents a new data-efficient approach to train neural operators via incorporating a pseudo physics-informed module which maps the solution u and its derivative to the source function f using a limited (u, f) pairs. The learned mapping is then used iteratively as part of training a data-driven neural operator. The paper tested the performance of the proposed model against two baselines, namely, FNO and DeepONet across a range of benchmarks and a real-world application. The method is shown to enhance the accuracy of neural operators particularly with limited data.

**Strengths:**

Paper is systematically written and easy to read. The idea presented is quite interesting, novel and an important one given the issue of lack of data and partially known physics one often encounters in real world applications.

The proposed model seems to notably improve the baseline models performance in limited data regimes despite marginally increasing the training time. The results presented sufficiently support the claims made by the authors.

The effectiveness of the proposed model was assessed in a real-world context scenario in fatigue modeling where no comprehensive PDE exists to fully describe the system. With the use of the pseduo-PI approach and sparse data, the proposed model is able to achieve accurate performance.

The author have ensured to highlight the limitations of the proposed models (e.g., being opaque and non-interpretable, and not applicable to input functions) which is appreciated.

**Weaknesses:**

The framework cannot be used for learning the mapping from the initial condition to the solution and the examples provided are mainly limited to mapping the source function to the solution.

It will help if the training procedure is described step by step, with one of the examples used in the results.

Citing and differentiating your work from this paper is recommended - https://www.sciencedirect.com/science/article/abs/pii/S0021999120307166

**Questions:**

Is it possible to use other operators (apart from differentiation) in the initial physics learning? For example, introducing operators such as sines, cosines or other complex ones? How about utilizing some concepts from this paper - https://arxiv.org/abs/2207.06240?

---

> ### Author Response · Authors · 2024-11-24
>
> We thank the reviewer for the comments. C: comments; R: response
>
> >C1: The framework cannot be used for learning the mapping from the initial condition to the solution and the examples provided are mainly limited to mapping the source function to the solution.
>
> R1: Thank you for your insightful comment. We acknowledge that our current framework is limited to mapping the source function to the solution and cannot yet handle mappings from the initial condition to the solution. Addressing this limitation is an important avenue for our future work, and we are actively exploring methods to extend our approach in this direction.
>
> >C2: Is it possible to use other operators (apart from differentiation) in the initial physics learning? For example, introducing operators such as sines, cosines or other complex ones? How about utilizing some concepts from this paper - https://arxiv.org/abs/2207.06240?
>
> R2: Thanks for the great suggestion. We will try those experiments in the future.

---

> > ### Comment · Reviewer_KpCq · 2024-11-28
> >
> > Thanks authors for your comments!
> > A couple of things requested are not answered in these comments. Urge authors to consider these -
> > 1. It will help if the training procedure is described step by step, with one of the examples used in the results.
> > 2. Citing and differentiating your work from this paper is recommended - https://www.sciencedirect.com/science/article/abs/pii/S0021999120307166
> >
> > However, overall satisfied with the work and will keep my score unchanged.

---

### Official Review · Reviewer_MPgk · 2024-11-03

**Soundness:** 2
**Presentation:** 3
**Contribution:** 2
**Rating:** 3
**Confidence:** 5

**Summary:**

This work attempts to provide regularization to deep operator network training by adding a "pseudo-physics" component when there is no knowledge of the PDE to inform training. A comprehensive experimental study with ablation is provided.

**Strengths:**

This is a method that attempts to provide physics regularization to the training of deep operator networks, which are usually trained only from data.

A comprehensive ablation study is provided.

**Weaknesses:**

This is a bootstrapping approach where one attempts to learn the physics and then use it to improve training of the operator network over a data-drive baseline. It is not clear how the pseudo-physics constraint helps achieve a better solution. This could happen simply by additional training of the operator network. There is no firm rationale for how this should work.

No comparison is made with the physics-informed neural operator using the correct physics. The authors' method should give a solution with accuracy between the data-driven and the physics-informed cases, but we do not know how much improvement is made unless we can see what the accuracy of the fully physics-informed operator network is.

There is some incorrect terminology (see Questions) and incorrect technical statements. For example, in Section 3.1, the authors state that the PDE solution can be obtained through integration of Green's function, but this is only true for linear PDEs.

The authors assess the additional number of parameters in their model, which is small, but nothing is said about the additional training and inference time incurred. The latter is important because there is a lot of iterative training and refinement in the proposed method.

**Questions:**

The approach to learn the physics resembles that in Section IV-B of Zhang et al. "Deep Learning and Symbolic Regression for
Discovering Parametric Equations". The authors should give that reference and compare their approach to theirs.

Why do the authors use the acronym "DONet" for "DeepONet"? The latter is the term widely used in the literature.

On page 2, the discretized versions of u and f aren't "collocation points". That refers to points where a PDE residual is minimized.

Still page 2, the efficiency of FNO does not reside in performing the convolution in the frequency domain, per se, but in learning the parameters in the frequency domain.

On page 3, it's not clear what the authors mean by having more data by decomposing the 128x128 input in 16,384 points. This is still the same amount of data.

On page 4, the authors say that the convolution layer is used to compensate for errors in the discretization of the derivatives. How?

---

> ### Author Response · Authors · 2024-11-24
>
> We thank the reviewer for the constructive comments. C: comments; R: response
>
> >C1: This is a bootstrapping approach where one attempts to learn the physics and then use it to improve training of the operator network over a data-drive baseline. It is not clear how the pseudo-physics constraint helps achieve a better solution. This could happen simply by additional training of the operator network. There is no firm rationale for how this should work.
>
> R1: We respectfully disagree with this assessment and believe it **misrepresents** the rigor and results of our work. In numerous experiments, our method has consistently demonstrated significant improvements over standard NO, as evidenced by clear margins and thorough statistical analysis (e.g., repeated experiments with standard deviations reported). Furthermore, we ensured that **all competing methods were trained with sufficient iterations to guarantee convergence and to extract their optimal performance**. The suggestion that our improvements might be attributed solely to "additional training of the operator network" overlooks these careful controls and the detailed analysis we presented in Section 5.1. We encourage the reviewer to revisit this section for a comprehensive explanation of our experimental settings and results.
>
> >C2: No comparison is made with the physics-informed neural operator using the correct physics. The authors' method should give a solution with accuracy between the data-driven and the physics-informed cases, but we do not know how much improvement is made unless we can see what the accuracy of the fully physics-informed operator network is.
>
> R2: We appreciate the reviewer's suggestion. While the primary focus of our work is to demonstrate the effectiveness of PPI-NO in scenarios where the underlying physics is unknown, we agree that comparing with a physics-informed neural operator (PI-NO) using correct physics is an interesting avenue. Below, we present the results of such a comparison on the **Poisson data**:
>
> |  Relative $L_2$ | Training size=5  |  10 | 20  |  30 |
> |---|---|---|---|---|
> | PI-NO with 200 generated samples| 0.1890(0.0042) | 0.0863(0.0020) | 0.0596(0.0168) | 0.0492(0.0003)|
> |  Ours    |  0.1437(0.0062)    | 0.0771(0.0018)   |  0.0544(0.0009)    |  0.0458(0.0003)  |
>
> As the training size increases, the relative $L_2$ errors of the two methods become very close. This convergence is likely due to two factors: (1) our alternative training framework inherently involves running more epochs in total compared to PI-NO, although in this case, we capped PI-NO training at 1000 epochs per sample for computational efficiency; and (2) our alternative training framework demonstrates clear performance gains when the training size is small (e.g., size = 5).
>
> >C3: The authors assess the additional number of parameters in their model, which is small, but nothing is said about the additional training and inference time incurred. The latter is important because there is a lot of iterative training and refinement in the proposed method.
>
> R3: We appreciate the reviewer's observation regarding training and inference time. Our method is primarily designed for **data-scarce** scenarios, where the focus is on achieving improved performance despite limited data, even at the expense of longer training times. In such cases, the relatively low training cost makes this trade-off worthwhile.
>
> Below, we provide the training times for different training sizes on two datasets:
>
> |  running time(min) | Training size=5  |  10 | 20  |  30 |
> |---|---|---|---|---|
> | darcy | 13 | 14 | 16 | 17 |
> |  advection    |  4    | 4   |  5   |  5  |
>
> The training time depends on both the number of iterations in the alternative training framework and the resolution of the data. For instance, the Advection dataset is faster to train as it uses a lower resolution (64x64), while the Darcy dataset has higher computational requirements due to its higher resolution.
>
> >C4: The approach to learn the physics resembles that in Section IV-B of Zhang et al. "Deep Learning and Symbolic Regression for Discovering Parametric Equations". The authors should give that reference and compare their approach to theirs.
>
> R4: Thanks for providing the reference. We would love to cite and discuss this work. However, we would like to highlight that our approach is fundamentally different in its goals and methodology. While their work focuses on symbolic discovery, our framework aims to establish a synergy between equation discovery and operator learning, where both processes mutually influence and enhance each other.
>
> Unlike Zhang et al., whose primary goal is symbolic regression, **our approach integrates physics discovery with operator learning to achieve better generalization and predictive accuracy under data-scarce conditions**. We believe this distinction sets our work apart and aligns with the unique challenges and objectives we address.

---

> ### Author Response · Authors · 2024-11-24
>
> >C5: On page 2, the discretized versions of u and f aren't "collocation points". That refers to points where a PDE residual is minimized.
>
> R5: To clarify, in our paper, the discretized versions of u and f refer to their representations on a grid of specified resolutions, such as 128x128 in 2D. These points are sampled as part of the grid, and while they are not "collocation points" in the strict sense of minimizing a PDE residual, we use the term in a broader sense to describe the grid points where the values of u and f are computed.
>
> >C6: Still page 2, the efficiency of FNO does not reside in performing the convolution in the frequency domain, per se, but in learning the parameters in the frequency domain.
>
> R6: We respectfully disagree with the reviewer's statement. The efficiency of FNO lies primarily in its use of FFT operations, which enable computationally efficient and fast global convolutions. Without FFT, performing function convolutions would be significantly more expensive.
>
> While learning parameters in the frequency domain is an important aspect of FNO, it also introduces a notable memory bottleneck due to the need to store complex-valued parameters. This trade-off between computational efficiency and memory usage is a critical consideration in the design of FNO, and we will revise the manuscript to clarify this point and address any potential misunderstanding.
>
> >C7: On page 3, it's not clear what the authors mean by having more data by decomposing the 128x128 input in 16,384 points. This is still the same amount of data.
>
> R7: To clarify, our approach focuses on constructing pointwise models. The 128x128 resolution represents the grid of input data, where each point may include additional information, such as derivatives (e.g., for u in the $\phi$-network).
>
> The key distinction lies in how the PDE equation is evaluated. In our framework, the PDE can be learned and evaluated independently at each grid point. This is in contrast to standard neural operator (NO) methods, which treat the entire
> 128x128 grid as a single entity. By focusing on pointwise evaluation, we simplify both the training process and the network design, making it more computationally efficient and easier to train.
>
> >C8: On page 4, the authors say that the convolution layer is used to compensate for errors in the discretization of the derivatives. How?
>
> R8: Thank you for raising this question. The convolution layer in the $\phi$-network is used to incorporate neighboring information into the pointwise model, which helps to compensate for discretization errors in the derivatives. Unlike using an MLP for purely pointwise modeling, the convolution layer effectively captures local spatial relationships between grid points, thereby improving the accuracy of the learned model. Cited by PDE-Net paper[1], the core idea is that the convolutional filters can be designed or learned to approximate finite difference operators.
>
> [1]PDE-Net: Learning PDEs from Data - https://arxiv.org/abs/1710.09668

---

> ### Comment · Reviewer_MPgk · 2024-11-24
>
> Here are my comments about the authors' replies:
>
> R2: As stated in the original comment, "The authors' method should give a solution with accuracy between the data-driven and the physics-informed cases". What this means is that if all methods use the same amount of data, the performance of the author's method should be in-between the performances of the other two. It is not expected that it would be superior to the physics-informed method, since the latter is using the true physics model. This is not what is observed in the new results reported by the authors.
>
> R3: The authors did not provide a comparison with the baseline data-driven method.
>
> R4: The authors cannot use the term collocation points in a "broader" sense. The term has had a well-defined meaning in the scientific computation literature long before the emergence of PINNs.
>
> R7: The author's reply does not address the question. Why is there more data?
>
> Overall, the author's replies were not satisfactory. I would like to keep my score.

---

### Comment · Area_Chair_c3AU · 2024-11-26

Dear all,

The deadline for the authors-reviewers phase is approaching (December 2).

@For reviewers, please read, acknowledge and possibly further discuss the authors' responses to your comments. While decisions do not need to be made at this stage, please make sure to reevaluate your score in light of the authors' responses and of the discussion.

- You can increase your score if you feel that the authors have addressed your concerns and the paper is now stronger.
- You can decrease your score if you have new concerns that have not been addressed by the authors.
- You can keep your score if you feel that the authors have not addressed your concerns or that remaining concerns are critical.

Importantly, you are not expected to update your score. Nevertheless, to reach fair and informed decisions, you should make sure that your score reflects the quality of the paper as you see it now. Your review (either positive or negative) should be based on factual arguments rather than opinions. In particular, if the authors have successfully answered most of your initial concerns, your score should reflect this, as it otherwise means that your initial score was not entirely grounded by the arguments you provided in your review. Ponder whether the paper makes valuable scientific contributions from which the ICLR community could benefit, over subjective preferences or unreasonable expectations.

@For authors, please respond to remaining concerns and questions raised by the reviewers. Make sure to provide short and clear answers. If needed, you can also update the PDF of the paper to reflect changes in the text. Please note however that reviewers are not expected to re-review the paper, so your response should ideally be self-contained.

The AC.

---

### Meta-Review · Area_Chair_c3AU · 2024-12-20

**Metareview:**

The reviewers are somewhat divided (3-6-3-5-6-3) about the paper, but they overall lean towards rejection. The paper introduces pseudo physics-informed neural operators to bypass the need for an accurate understanding of the target physical system. The approach is well-motivated, but the reviewers have raised a number of concerns about the clarity, the presentation, and the evaluation of the approach. The author-reviewer discussion has been constructive and has led to a number of clarifications and improvements, with the addition of new results. However, the reviewers still believe further work is needed to improve the evaluation. For these reasons, I recommend rejection. I encourage the authors to address the reviewers' comments and to resubmit to a future conference.

**Additional Comments On Reviewer Discussion:**

The author-reviewer discussion has been constructive and has led to a number of clarifications and improvements, with the addition of new results.

---

### Decision · Program_Chairs · 2025-01-22

Reject